

# Nonreciprocal superconducting transport and the spin Hall effect in gyrotropic structures

Tim Kokkeler[1,2], Ilya Tokatly[3,4] and F. Sebastian Bergeret[1,5]

**1** Donostia International Physics Center (DIPC), 20018 Donostia–San Sebastián, Spain
**2** University of Twente, 7522 NB Enschede, The Netherlands
**3** Nano-Bio Spectroscopy Group, Departamento de Polímeros y Materiales Avanzados,
Universidad del País Vasco, 20018 Donostia-San Sebastián, Spain
**4** IKERBASQUE, Basque Foundation for Science, 48009 Bilbao, Spain
**5** Centro de Física de Materiales (CFM-MPC) Centro Mixto CSIC-UPV/EHU,
E-20018 Donostia-San Sebastián, Spain

⋆ tim.kokkeler@dipc.org , † ilya.tokatly@ehu.es , ‡ fs.bergeret@csic.es

## Abstract

The search for superconducting systems exhibiting nonreciprocal transport and, specifically, the diode effect, has proliferated in recent years. This trend has encompassed a wide variety of systems, including planar hybrid structures, asymmetric SQUIDs, and certain noncentrosymmetric superconductors. A common feature of such systems is a gyrotropic symmetry, realized on different scales and characterized by a polar vector. Alongside time-reversal symmetry breaking, the presence of a polar axis allows for magnetoelectric effects, which, when combined with proximity-induced superconductivity, results in spontaneous non-dissipative currents that underpin the superconducting diode effect. With this symmetry established, we present a comprehensive theoretical study of transport in a lateral Josephson junction composed of a normal metal supporting the spin Hall effect, and attached to a ferromagnetic insulator. Due to the presence of the latter, magnetoelectric effects arise without requiring external magnetic fields. We determine the dependence of the anomalous currents on the spin relaxation length and the transport parameters commonly used in spintronics to characterize the interface between the metal and the ferromagnetic insulator. Therefore, our theory naturally unifies nonreciprocal transport in superconducting systems with classical spintronic effects, such as the spin Hall effect, spin galvanic effect, and spin Hall magnetoresistance. We propose an experiment involving measurements of magnetoresistance in the normal state and nonreciprocal transport in the superconducting state. Such experiment would, on the one hand, allow for determining the parameters of the model and thus verifying with a greater precision the theories of magnetoelectric effects in normal systems. On the other hand, it would contribute to a deeper understanding of the underlying microscopic origins that determine these parameters.



## Contents

## 1 Introduction

Recently, there has been a significant surge of attention directed towards nonreciprocal transport within superconducting structures. This heightened interest has been particularly focused on the investigation of the superconducting diode effect (SDE) [1–22, 22–42], as well as the study of spontaneous supercurrents, referred to as anomalous currents [43–57].

Both of these phenomena share a common origin and stem from the spin-galvanic effect (SGE) [48], also known in the literature as the (inverse) Edelstein effect [48,58–64]. The SGE involves the generation of charge currents from a spin accumulation and has been extensively studied in non-superconducting systems. In the normal state, due to gauge invariance, the SGE entails generating a charge current only from a nonequilibrium spin distribution, e. g. from a steady spin accumulation induced by a time-dependent magnetic field [64]. In contrast, in the superconducting state, gauge invariance does not prevent the generation of finite supercurrents from a static equilibrium spin polarization. This phenomenon is the superconducting, dissipationless, counterpart of the SGE [48]. In Josephson junctions, it manifests through the emergence of an anomalous phase denoted as $\phi_0$, which has been observed in numerous experiments. If the current-phase relation of the junction contains second or higher harmonics, $\phi_0$-junctions may demonstrate a critical current whose magnitude depends on the direction of the applied current, which is known as the Josephson SDE [65].

Thus, the SGE in normal systems, anomalous currents, $\phi_0$-junctions, and the SDE all manifest as different facets of the same phenomenon. In this introductory section, we collectively refer to these manifestations as the SGE.

The SGE may appear in systems where time-reversal and inversion symmetries are broken. Time-reversal symmetry can be broken by an external magnetic field or through the use of ferromagnets. Meanwhile, it is a specific type of breaking the inversion symmetry that leads to the emergence of the SGE, and consequently to anomalous currents and the SDE in the superconducting state.

Formally, the appearance of magnetoelectric effects can be understood by considering the possible tensors in a system. Spin, as a pseudovector $s_i$, with $i = x, y, z$, maintains its direction

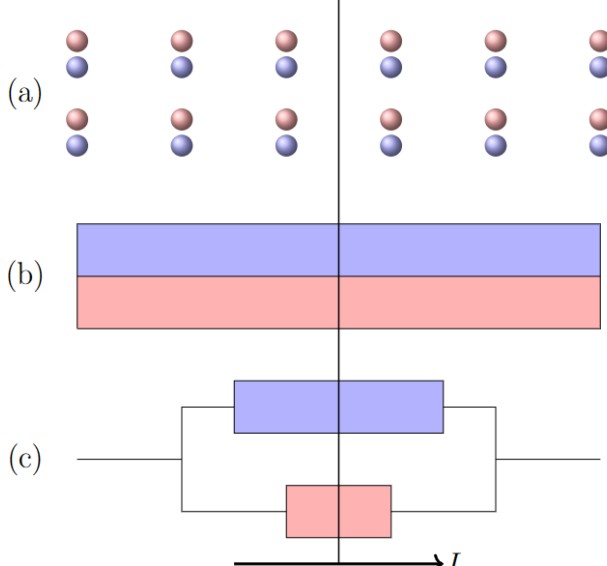

Figure 1: Illustration of generation of gyrotropy in a system via a polar vector on different scales. The (vertical) polar vector can be generated on a microscopic level in crystals with broken inversion symmetry (a), or via the junction geometry. This can be done on a mesoscopic level via the interface between two materials (b), or even in a macroscopic scale in devices such as in asymmetric SQUIDs (c). All these systems, when subject to time reversal breaking, allow for magnetoelectric and nonreciprocal transport effects.

upon inversion of the system. Therefore, in order for spin to be coupled to the supercurrent $J_i$–a polar vector–it requires a connection through a second-rank pseudotensor [48, 59]:

$$J_i = \alpha_{il} s_l \,. \tag{1}$$

This relation governs the interaction of spins or magnetic fields with electric current and is essential for both the SGE in normal metals and non-reciprocal effects in superconductors. A well-known example of systems satisfying Eq. (1) are two-dimensional Rashba conductors [26, 66–68]. These materials are characterized by a polar vector **n** perpendicular to the conducting plane. Thus, the pseudotensor in (1) can be constructed through the contraction of the fully antisymmetric pseudotensor, $\varepsilon_{ijk}$ and the components of **n**: $\alpha_{il} = \varepsilon_{ilk} n_k$.

   In general, a second rank pseudotensor is allowed in bulk materials with a so call gyrotropic crystal structure, widely studied in the context of crystal optics [69] from where the terminology is originating. The gyrotropy, underlying natural optical activity and the SGE, implies a lack of inversion symmetry. However, it is important to emphasize that not all non-centrosymmetric crystals exhibit gyrotropy. For example, zink blend materials, such as GaAs, are not gyrotropic, even though they are non-centrosymmetric. In bulk crystals, the SGE is allowed only within 18 out of in total 21 non-centrosymmetric classes [69, 70], the so-called gyrotropic classes. In infinite systems, gyrotropy is fully determined by the crystal structure of the material, as illustrated in Fig. 1(a). All gyrotropic materials support non-reciprocal transport if time-reversal symmetry is broken, for example by a magnetic field. Examples of gyrotropic materials that are currently used in proposals and realizations for superconducting diodes are, apart from the Rashba superconductors, $MoTe_2$ and $WeTe_2$ [71, 72].

   In this respect the SGE and non-reciprocal transport are fundamentally different from the spin Hall effect (SHE), which is known to be present even in isotropic materials [60, 73–81].

Indeed, the spin-Hall effect describes the interconversion of electrical current with spin current instead of spin. The spin current is a second rank pseudotensor, $J_{ik}$, with indices for both spin polarization and the direction of the current flow. To couple the spin current with the charge current $J_l$, which transforms as a vector, a third rank pseudotensor $\chi_{ikl}$ is needed:

$$J_{ik} = \chi_{ikl} J_l \,. \tag{2}$$

Even in isotropic materials, there exists a natural third rank pseudotensor, $\chi_{ikl} = \theta \epsilon_{ikl}$, where $\epsilon_{ikl}$ is the fully antisymmetric tensor and $\theta$ is a scalar, usually called the spin-Hall angle. Consequently, $\epsilon_{ikl}$ is sufficient to establish a connection between the charge current and a perpendicular spin current, even in materials with centrosymmetric crystal structures. This explains why the SHE is allowed in any material and, in infinite systems, is not related to SGE or non-reciprocal transport.

On the other hand, in finite systems, gyrotropy can exist even when the constituent materials possess centrosymmetric crystal structures. For example, near the edge of a sample the symmetry of the crystal structure is broken on a microscopic level, which may lead to the formation of conducting surface states, such as in topological insulators [82,83] or interfaces with Rashba-like surface bands [84–86]. These two situations correspond to effective two-dimensional gyrotropic structures at hybrid interfaces.

Gyrotropy can also arise at other scales. For example, it can originate from the specific design of a macroscopic device itself. An example of this is asymmetric SQUIDs, which have been widely explored in the context of the SDE [24, 28, 41, 42, 65] (see Fig. 1(c)). Moreover, in a hybrid bilayer system, reflection symmetry is broken as there is a polar axis perpendicular to the hybrid interface (see Fig. 1(b)). In this case, gyrotropy is defined at the mesoscopic transport scale by the mere existence of an interface, rather than by microscopic symmetry breaking at this edge. This constitutes the central situation analyzed in the present work.

A significant advantage of this structure or any lateral structure as the one shown in Fig. 2(a) is that they do not require specific assumptions about either the crystal structure or the interface. The mere presence of hybrid interfaces implies a broken reflection symmetry and indicates the existence of a polar vector perpendicular to that edge, hence the system is gyrotropic and supports SGE. In normal hybrid structures containing interfaces between materials with substantial bulk SOC, a spin-to-charge conversion due to SGE has been a topic of extensive studies, both theoretically [64, 87–89] and experimentally [90–98]. From the symmetry perspective it is irrelevant whether the SGE is mediated by SOC in the bulk of materials, comes from the surface Rashba band, or originates from the interfacial SOC and spin-flip scattering off the interface. In such lateral structures, even for trivial spin-inert interfaces without substantial interfacial SOC, the bulk SOC in the form of SHE generates SGE at the scale of spin diffusion length. The well known example in the normal spintronics is the current-induced edge spin accumulation caused by the bulk SHE. The superconducting analogue of such mesoscopic SGE is our main concern here.

In this work, we focus on superconducting lateral structures and address the previously unstudied situation in which nonreciprocal transport effects arise from spin-charge interconversion in the bulk of a centrosymmetric material. Specifically, we analyze the simplest structure that may exhibit the SGE in the absence of an external field: lateral structures, as the one shown in Fig. 2(a). In these systems, gyrotropy emerges at a mesoscopic scale determined by the spin diffusion and superconducting coherence lengths. We present a theoretical framework that unifies and generalizes two well-established theories: Superconducting Proximity Effect and charge-spin conversion in Spin Hall systems, in particular the spin Hall magnetoresistance phenomena [99]. We show how results in one of these fields may be used to predict phenomena in the other, seemingly disparate, field.

Concretely, the system under consideration in this paper consists of a normal metal (N) with inversion symmetric crystal structure and bulk SOC, sandwiched between a superconductor (SC) and a ferromagnetic insulator (FI). This configuration can be either realized in a Josephson configuration (Fig. 2(a)) or as a continuous structure (Fig. 2(b)). While the former coincides with a real setup that has recently been explored experimentally in Ref. [100], the latter is a geometry in which the diode effect can be explored analytically.

Let us focus on the geometry shown in Figs 2. While superconducting singlet correlations are induced in the normal region via the superconducting proximity effect, the FI introduces an interfacial exchange field. This field converts a part of the singlet pairs into triplets [101]. By drawing an analogy between singlet/triplet and charge/spin [48, 101, 102], one can envision the FI interface as an injector generating a triplet (spin) accumulation at the interface with the SC. These triplets can then diffuse in the vertical direction, generating a supercurrent parallel to the interface through the SOC. This spontaneous current leads to the anomalous phase in finite structures, as our calculations below demonstrate.

In order to describe transport through the proximitized normal metal, we use the well-established Usadel equation generalized for the case of extrinsic spin-orbit coupling [103,104]. This equation is complemented by boundary conditions at the interface. For the S/N interface, we apply the well-known Kupryanov-Lukichev boundary condition [105], whereas to describe the N/FI interface we establish boundary conditions based on symmetry arguments. At this interface we identify two types of terms: one describing the interfacial exchange coupling, quantifiable using the imaginary part of the so-called spin-mixing conductance $G_i$ [99, 106]. The second contribution from the boundary condition takes on the form of a spin-relaxation tensor in a magnet with uniaxial symmetry [107]. Following convention, interfacial spin-relaxation can be characterized by two further parameters: the so-called spin-sink conductance $G_s$ [108] and the real part of the spin-mixing conductance $G_r$. The three parameters $G_{i,r,s}$ are widely used in spintronics for normal metals to describe hybrid interfaces.

One important point of concern is the strong spin relaxation in metals exhibiting SOC [109–111]. In general, two main sources of spin relaxation exist. One mechanism arises due to SOC combined with disorder. Since SOC preserves time-reversal symmetry, the spin-orbit relaxation solely impacts triplet correlations, leaving singlets unaffected. On the other hand, the presence of the ferromagnetic insulator (FI) can lead to another form of spin relaxation, due to magnetic disorder. The latter does suppress superconductivity [112] and consequently, it suppresses supercurrents and the Josephson coupling as well. We show that non-reciprocal transport in lateral junctions is robust against bulk spin-relaxation due to SOC, as long as the thickness $d$ of the normal metal is sufficiently small, but is severely affected by the magnetic disorder at the interface.

The structure of the article is as follows. In Sec. 2 we present our model and the equations governing transport in dirty normal metals proximitized by superconductors and ferromagnetic insulators, including the spin Hall angle. In Sec. 3 we use the linearized version of these equations to obtain the pair amplitudes in a Josephson junction, show the existence of a $\phi_0$-effect and identify conditions to maximize this effect. To illustrate the existence of a diode effect within the linear approach, we use the same set of equations to describe a different geometry in Sec. 4. We show that in this setup the maximum of the magnitude of the supercurrent is different in opposite directions, even after linearization. Finally, in Sec. 5, we summarize our results, and propose an experiment based on them.

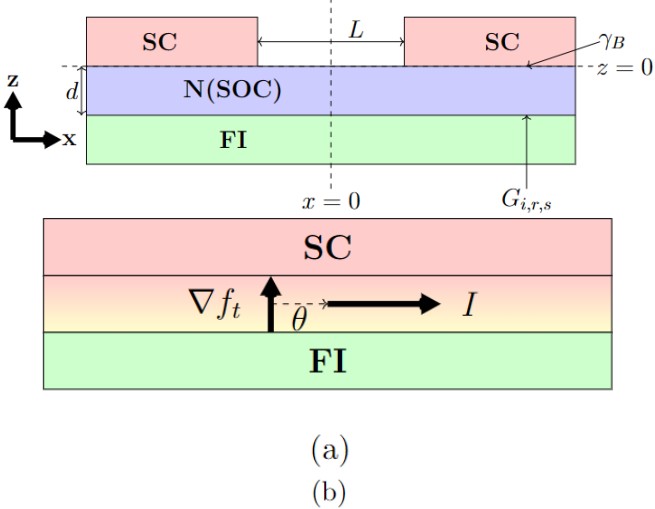

(a)

(b)

Figure 2: *(a)* Lateral Josephson junction proximized by a ferromagnetic insulator from below. The normal metal N exhibits a sizable spin-orbit coupling (SOC). The interface between the N and the superconductor leads (SC) is characterized by the parameter $\gamma_B$. The interface between N and ferromagnetic insulator (FI) is characterized by the spin mixing conductances $G_{r,i}$. *(b)* Schematic illustration of the generation of an anomalous current via the combination of the spin Hall effect (SHE) and the presence of a magnetic interface. The superconductor induces singlet pair correlations (red) into the metal, which are converted into a mixture of singlet and triplet pair correlations (yellow) by the ferromagnetic insulator. The gradient of the triplet correlations causes a vertical diffusive spin current, which in turn causes a charge current via the SHE.

## 2 The model

We consider a normal metal with spin-orbit coupling. The normal metal extends infinitely in both x and y-directions, but has a finite thickness $d$ in the z-direction. It is proximitized by a ferromagnetic insulator (FI) from below and by conventional superconductors (SC) from the top, placed only for $|x| > L/2$. A schematic of the setup is shown in Fig. 2(a). The system has translational invariance in the $y$-direction, but the dependence on both x and z coordinates is non-trivial.

We focus here on a metallic N region with a Fermi energy much larger than any other energy scale in the problem, allowing the use of the quasiclassical formalism [113,114]. Additionally, we assume the metal to be dirty, that is, that the scattering rate $\tau^{-1}$ is larger than any other energy scale except the Fermi energy.

We assume that the metal possesses inversion symmetry, and the spin-orbit coupling (SOC) within it leads to spin relaxation and the spin Hall effect (SHE). Since the system is in the diffusive limit, it can be described using the momentum-independent quasiclassical Green's functions that satisfy the Usadel equation [115]. A concise and useful way to express the Usadel equation is through a variational principle, achieved by an effective action, commonly used to formulate a nonlinear $\sigma$-model for disordered systems [116]. The effective action, which, at the saddle point, generates the Usadel equation in a material with SOC supporting the spin Hall effect, was derived in Ref. [103]:

$$S = \frac{i\pi}{8}\mathrm{Tr}\left( D(\nabla \breve{g})^2 + 4i\varepsilon\tau_3\breve{g} + D\theta\epsilon_{ijk}\sigma_k\breve{g}\partial_i\breve{g}\partial_j\breve{g} - iD\chi\epsilon_{ijk}\sigma_k\partial_i\breve{g}\partial_j\breve{g} - \frac{1}{\tau_{so}}\breve{g}\sigma_k\breve{g}\sigma_k \right). \quad (3)$$

Here $\check{g}$ is the quasiclassical Green's function which satisfies the normalization condition $\check{g}^2 = \check{\mathbf{1}}$, where $\check{\mathbf{1}}$ denotes the unit matrix in Nambu⊗spin space.[1] It is a 4×4 matrix in the Nambu⊗spin space. The matrices $\sigma_a(\tau_a)$, $a = x, y, z$ are the Pauli matrices in spin (Nambu) space. Summation over repeated indices is implied and $D$ is the diffusion constant of the normal metal. The second order term in spin Pauli-matrices described the spin-relaxation, with $\tau_{so}$ being the spin-relaxation time. There are two linear terms in $\sigma$'s: The spin Hall term, proportional to the spin Hall angle $\theta$, which describes the charge-spin conversion, and the so-called spin current swapping term, proportional to spin swapping coefficient $\chi$. Due to the normalization condition $\check{g}^2 = \check{\mathbf{1}}$ these two terms are the only allowed first order terms. While, as explained later, the spin current swapping effect is not relevant for the present work, it has been shown in Ref. [117] that by symmetry it always accompanies the spin Hall effect, and we therefore include the corresponding term in the effective action of Eq. (3) for generality.

It is worth noting that our theory captures all mechanisms (for example, coming from intrinsic or extrinsic SOC) of spin-charge coupling and spin relaxation, allowed by symmetry in isotropic systems. Different types of SOC generally lead to different relations between the transport coefficients $\theta, \chi, \frac{1}{\tau_{so}}$ [103,118]. Therefore, in our theory, without specifying a microscopic origin of $\theta$, $\chi$ and $1/\tau_{so}$, we treat them as independent parameters. Because isotropic systems are not gyrotropic, the action in Eq. (3) does not have any coupling between the charge current and the spin and hence it does not support the bulk superconducting diode effect.

The gyrotropy therefore has to be introduced using a boundary. To describe the boundaries at $z = 0$ with the superconductor and at $z = d$ with the FI we introduce a boundary action, also up to second order in the spin Pauli matrices:

$$S_b = \frac{i\pi}{8\sigma_N} \text{Tr}\Big(G_i \boldsymbol{m} \cdot \boldsymbol{\sigma} \tau_3 \check{g} - G_r \boldsymbol{m} \cdot \boldsymbol{\sigma} \tau_3 \check{g} \boldsymbol{m} \cdot \boldsymbol{\sigma} \tau_3 \check{g} - G_s \sigma_k \tau_3 \check{g} \sigma_k \tau_3 \check{g}\Big)_{z=-d} + \frac{i\pi}{8\sigma_N R_\square} \text{Tr}\Big(\check{g}_s \check{g}\Big)_{z=0}.$$
(4)

The last term generates the standard Kuprianov-Lukichev boundary conditions [105] and describes the interface with the SCs. $R_\square$ is the SC/N interface resistance, $\sigma_N$ is the conductivity of the N metal. The first three terms in Eq. (4) describe the FI/N interface, with $\boldsymbol{m}$ being the direction of magnetization in the FI. They are constructed by considering all local terms allowed by symmetry, up to second order in the spin Pauli matrices, in the presence of the unit pseudovector $\boldsymbol{m}$. This boundary functional describes the gyrotropy of the interface, enabling non-reciprocal transport, as the terms at $z = 0$ differ from those at $z = -d$.

As the interaction with the FI is assumed to occur via an exchange field, breaking time reversal symmetry, the Pauli matrices in spin space, $\sigma_k$, are necessarily accompanied by a $\tau_3$ in Nambu space. The first term in Eq. (4) corresponds to the direct exchange interaction term, which is the first order in Pauli matrices. Following the customary convention in spintronics, this term is characterized by the interfacial conductance $G_i$ [99, 106]. The second and third terms are of second order in the Pauli matrices and arise from the spin-relaxation tensor, which, in the case of uniaxial symmetry, has the following general structure, $\Gamma_{ij} = (1/\tau_\perp)\delta_{ij} + (1/\tau_\parallel - 1/\tau_\perp)m_i m_j$, where $\tau_\parallel$ and $\tau_\perp$ are the longitudinal and transverse spin relaxation times. These terms can be expressed conventionally [107] by identifying $G_s = e^2 \nu_0/\tau_\parallel$ and $G_r = e^2 \nu_0(1/\tau_\perp - 1/\tau_\parallel)$, with $\nu_0$ being the normal density of states at the Fermi level.

It is worth noting that, in principle, similar boundary terms at hybrid interfaces can be formulated due to the existence of the polar vector $\boldsymbol{n}$ perpendicular to the interface. The influence of spin-orbit coupling in this scenario also induces spin relaxation. In such cases,

---

[1]The effective action, Eq. (3), is typically formulated in terms of the matrix field $\check{Q}$, which arises following a Hubbard-Stratonovich transformation [131], and satisfies $\check{Q}^2 = 1$. Instead, we have directly introduced the quasiclassical Green's function $\check{g}$ to reflect our focus on the saddle point equation, i.e. the Usadel equation.

the boundary action adopts the same form as Eq. (4), albeit without the $\tau_3$ matrices. This difference only leads to a renormalization of coefficients in the normal state. However, in the superconducting case, the above two situations yield distinct consequences: whereas magnetic impurities suppress both singlet and triplet correlations, those arising from spin-orbit coupling keep the singlet component unaffected. In the subsequent discussion, we exclusively consider magnetic interactions at the interfaces, though we will get back to this aspect later.

The saddle-point equation for the action Eq. (3) is the Usadel equation, which was derived in Ref. [103] (see Eq. (21) in that work). Here, aiming at the subsequent linearization (see Sec. 3), we present a simplified form of it in the Matsubara representation, where $i\varepsilon \rightarrow \omega = (2n+1)\pi T$, $T$ is the temperature, and assuming that $\theta$ and $\chi$ are both constant in the N region

$$-\partial_a \check{J}_a = [\omega \tau_3, \check{g}] + \frac{1}{4\tau_{\text{so}}}[\sigma_a \check{g}\sigma_a, \check{g}],\tag{5}$$

where the matrix current $\hat{J}_a$ is given by:

$$\check{J}_a = -D\check{g}\partial_a \check{g} + \epsilon_{akj}\theta D\{\partial_k \check{g}, \sigma_j\} + i\epsilon_{akj}\chi D[\check{g}\partial_k \check{g}, \sigma_j],\tag{6}$$

where $\{,\}$ denotes the matrix anticommutator. The boundary conditions at the interfaces with the SC and FI can be obtained from Eqs. (3,4). They can be written in terms of the currents flowing through the boundaries as

$$-\check{J}_z(z=0)/D = \frac{1}{\sigma_N R_\square}[\check{g}, \check{g}_s],\tag{7}$$

and

$$-\check{J}_z(z=-d)/D = i\frac{G_i}{\sigma_N}[\boldsymbol{m}\cdot\boldsymbol{\sigma}\tau_3, \check{g}] + \frac{G_s}{\sigma_N}[\sigma_k\tau_3\check{g}\tau_3\sigma_k, \check{g}] + \frac{G_r}{\sigma_N}[\boldsymbol{m}\cdot\boldsymbol{\sigma}\tau_3\check{g}\boldsymbol{m}\cdot\boldsymbol{\sigma}\tau_3, \check{g}].\tag{8}$$

At the interface with the superconductor, the obtained boundary condition Eq. (7) coincides with the Kuprianov-Luckichev boundary condition [105].

At the N/FI interface, one can easily check that Eq. (8), in the normal state corresponds to the boundary condition between a normal metal and a ferromagnetic insulator [106], widely used in the theory of the spin Hall magnetoresistance (SMR) [99, 107]. In the superconducting state, Eq. (8) can be related to the boundary conditions for interfaces with a spin-mixing angle derived in Ref. [119], up to second order in Pauli matrices.

The Usadel equation, Eq. (5), along with the boundary conditions Eqs. (7-8) and the normalization condition $\check{g}^2 = \check{\mathbf{1}}$, defines the boundary problem for the quasiclassical matrix Green's function $\check{g}$. Once this matrix Green's function $\check{g}$ is obtained, the current can be computed through Eq. (6) and to explore the transport properties of the junction. In the next section, we employ these equations to calculate the anomalous current and to understand its origin.

## 3 The anomalous current

In this section, we focus on calculating the anomalous current in the lateral Josephson junction shown in Fig. 2. The N/FI interface defines a polar vector $\boldsymbol{n}$, the normal to this interface. Therefore, the spin-galvanic effect is allowed by symmetry. According to Eq. (1), a supercurrent can be generated, satisfying $\boldsymbol{j} \propto \boldsymbol{n} \times \boldsymbol{m}$, where $\boldsymbol{m}$ is the unit vector indicating the direction of the magnetization of the FI. In the structure depicted in Fig. 2, $\boldsymbol{n} = \hat{\boldsymbol{z}}$, and thus, to maximize the spontaneous current in the x-direction, we set $\boldsymbol{m} = \hat{\boldsymbol{y}}$. In this geometry $\sigma_{x,z}$

do not appear in the problem and hence the Green's function necessarily satisfies $\check{g} = \sigma_y \check{g} \sigma_y$. As a result, the spin-swapping coefficient naturally drops out of the equation.

To obtain analytical results for the anomalous current, we linearized the boundary problem described by Eqs (5-8). This is justified by assuming that the contact between the superconductor and the normal metal is weak, resulting in a weak superconducting proximity effect. Under this condition, the pair amplitudes in the normal region, which correspond to the amplitude of the anomalous Green's functions, are small. Thus, the Green's function can be approximated as $\check{g} \approx \text{sign}(\omega)\tau_3\sigma_0 + \hat{F}_s\sigma_0 + \hat{F}_t\sigma_y\text{sign}(\omega)$ where the singlet (s) and triplet (t) anomalous parts are parameterized as $\hat{F}_{s,t} = \text{Re}(F_s)\tau_1 + \text{Im}(F_s)\tau_2$ and $\hat{F}_t = \text{Re}(F_t)\tau_2 + \text{Im}(F_t)\tau_1$, where $F_{s,t}$ are the singlet and triplet pair amplitudes. Since singlet(triplet) correlations are even(odd) in frequency this parameterization ensures that both $\hat{F}_s$ and $\hat{F}_t$ are even-frequency.

The current is given by

$$I = \frac{2\pi\sigma_N}{e}T\sum_n j(\omega), \tag{9}$$

where the summation is over the Matsubara frequencies. It is convenient to calculate the spectral current $j(\omega)$, at the interface between the normal metal and one of the electrodes by using the boundary condition Eq. (7). As elaborated upon in appendix A, the supercurrent flowing from the left (L) to the right (R) electrodes is expressed in terms of the kernel $Q$ of the linearized problem as

$$j(\omega) = \gamma_B^2 \text{Im}\left\{\int_L dx \int_R dx' f_L^* Q(x-x')f_R\right\}, \tag{10}$$

where $\gamma_B = \frac{1}{\sigma_N R_\square}$, $f_{L,R}$ are the BCS pair amplitudes at the left (right) electrodes, that is,

$$f_{s0}(x) = f_L\Theta\left(-\frac{L}{2}-x\right) + f_R\Theta\left(x-\frac{L}{2}\right), \tag{11}$$

$$f_{L,R} = \frac{\Delta}{\sqrt{\omega^2+\Delta^2}}e^{i\phi_{L,R}}, \tag{12}$$

where $\Delta$ is the pair potential of the superconductor and $\phi_{L,R}$ are the phases of the superconductors, with $\phi_R = -\phi_L = \phi$. $Q(x-x')$ is the kernel defining the linear relation between the anomalous functions at a contact surface.

We assume that the junction is translation invariant in the $y$-direction. In this case the anomalous functions are independent of $y$, and the problem reduces to two dimensions. It is convenient to Fourier transform the anomalous functions over the $x$-direction and to express all quantities as a function of Fourier momentum $k$, while keeping the $z$-coordinate in position space. In momentum space the linear relation between the pair potential in the normal metal $F_s(k,z=0)$ and the pair potential in the superconductor $f_{s0}(k)$ via the kernel $Q$ takes the form:

$$F_s(k,z=0) = \gamma_B Q(k)f_{s0}(k), \tag{13}$$

where $Q(k)$ is the Fourier transform of $Q(x-x')$.

The singlet $F_s(k,z)$, and triplet $F_t(k,z)$ components of the condensate satisfy the linearized Usadel equation obtained by linearizing Eqs. (5-6) and Fourier transforming over the $x$-direction:

$$\begin{aligned}\partial_z^2 F_s(k,z) - (k^2+\kappa_s^2)F_s(k,z) &= 0, \\ \partial_z^2 F_t(k,z) - (k^2+\kappa_t^2)F_t(k,z) &= 0.\end{aligned} \tag{14}$$

These are two decoupled equations that determine the characteristic inverse lengths over which the singlet and triplet correlations vary: $\kappa_s^2 = \frac{2|\omega|}{D}$ and $\kappa_t^2 = \kappa_s^2 + \frac{1}{l_{so}^2}$, where $l_{so} = v_F\tau_{so}$

represents the spin relaxation length due to bulk SOC. As expected, because SOC preserves time reversal symmetry, it does not affect the singlet component of the condensate, whereas it has a detrimental effect on the triplet one.

The full boundary problem is defined by Eqs. (14) and the linearized boundary conditions obtained Fourier transforming Eq. (7) at the interface with the superconductor ($z = 0$), and Eq. (8) at the interface with the ferromagnetic insulator ($z = -d$):

$$
\begin{aligned}
&\partial_z F_s(k,-d) - \gamma_{ik} F_t(k,-d) - (\gamma_r + 3\gamma_s) F_s(k,-d) = 0, \\
&\partial_z F_t(k,-d) + \gamma_{ik} F_s(k,-d) - (\gamma_r + \gamma_s) F_t(k,-d) = 0, \\
&\partial_z F_s(k,0) + k\theta F_t(k,0) = \gamma_B f_{s0}(k), \\
&\partial_z F_t(k,0) - k\theta F_s(k,0) = 0,
\end{aligned}
\tag{15}
$$

where $\gamma_{ik} = \gamma_i - k\theta$, $\gamma_i = \frac{G_i}{\sigma_N}$, $\gamma_r = \frac{G_r}{\sigma_N}$, and $\gamma_s = \frac{G_s}{\sigma_N}$. Some remarks can be made at this stage: First, the singlet and triplet components satisfying Eqs. (14) are coupled via the $\gamma_i$ term. The latter describes an interfacial exchange field and, as in ferromagnets, it is the source of singlet-triplet conversion [120]. Both $\gamma_r$ and $\gamma_s$ act as pair-breaking mechanisms similar to magnetic impurities [112], thus suppressing both singlet and triplet correlations near the interface. Finally, the spin Hall angle only enters the boundary problem through the boundary conditions, as given by the last two Eqs. (15).

The appearance of the anomalous current, and hence the finite $\phi_0$, can be explained as follows (see Fig. 2b): The singlet component of the condensate is induced in the normal region via the proximity effect, third equation in (15). The singlet components diffuse in the normal metal over the length $\kappa_s^{-1}$, according to first Eq. (14). At the boundary with the FI the singlets are converted into triplet due to the interfacial exchange field, $\gamma_i$-term in first line of Eqs. (15). Triplet components diffuse in the normal region over distances of the order $\kappa_t^{-1}$, Eq. (14), generating a diffusive spin current in $z$-direction, which under the SHE, quantified by the SH angle $\theta$, it converts to a charge current in $x$-direction.

If the thickness is large, $\kappa_s d > 1$, the interfaces are far away and only a small portion of the singlet condensate reaches the FI interface. In this case the two proximity effects are almost decoupled and the anomalous current is negligibly small. Here we focus in the more interesting case in which the normal metal is thin enough that it is fully proximitized by the superconductor, that is, $\kappa_s d \ll 1$. Below we solve the above boundary problem defined by Eqs. (14), (15), determining the singlet amplitude $F_s$ and compute the kernel $Q(k)$ from Eq. (13). Technical details of the calculation are presented in Appendix A. In this thin N limit the kernel reads

$$
Q(k) = \frac{d(k^2 + \kappa_t^2)(1 + \gamma_r d)S + (\gamma_i^2 - 2\gamma_i\theta k)C + \gamma_r(1 + \gamma_r d)C}{((k^2 + \kappa_s^2)(k^2 + \kappa_t^2)d^2 S + (\gamma_i^2 - 2\gamma_i\theta k)C + 2\gamma_i\theta k + (k^2 + \kappa_t^2)\gamma_r dS + (k^2 + \kappa_s^2)\gamma_r dC + \gamma_r^2 C)},
\tag{16}
$$

where $C = \cosh \kappa_t d$ and $S = \frac{\sinh \kappa_t d}{\kappa_t d}$ were introduced for brevity of notation. To maximize the $\phi_0$-effect we consider first the limit of weak spin relaxation at the boundary, i.e. $\gamma_r, \gamma_s \ll \gamma_i$, which may correspond to certain Eu chalcogenide insulators [121]. Later, in Sec. 3.3, we discuss the influence of $\gamma_r$ and show how it limits nonreciprocal transport. We keep $l_{so}$ finite to show that anomalous currents may exist even if $L \gg l_{so}$. Because the numerator and denominator of Eq. (16) are analytic functions, $Q(k)$ is a rational function and hence the Fourier transform is calculated by summing over the residues at the poles. Poles with large imaginary part have exponentially suppressed residues due to the exponential factor $e^{ikL}$ that appears after the Fourier transform, as discussed in appendix A, see Eq. (A.6). Therefore, later on, we only take into account those poles with small imaginary part, $kd \lesssim 1$.

The current is computed from Eq. (10), which can be written in terms of the kernel and the phase difference $\phi$ between the electrodes as

$$j(\omega) = \gamma_B^2 \frac{\Delta^2}{\omega^2 + \Delta^2} \left[ \sin\phi \int_L dx \int_R dx' \operatorname{Re} Q(x - x') + \cos\phi \int_L dx \int_R dx' \operatorname{Im} Q(x - x') \right], \tag{17}$$

where $Q(x - x')$ is the inverse Fourier transform of Eq. (16). Since we use the linearized equation the current contains only first harmonics, $I(\phi) = I_0 \sin\phi + I_1 \cos\phi = I_c \sin(\phi - \phi_0)$, where $\phi_0$ is determined by

$$-\phi_0 = \tan^{-1} \frac{\sum_n \int_L dx \int_R dx' \operatorname{Im} Q(x - x')}{\sum_n \int_L dx \int_R dx' \operatorname{Re} Q(x - x')}. \tag{18}$$

Thus, if $\operatorname{Im} Q(x-x')$ is nonzero, the current $I(\phi)$ has an anomalous $\cos\phi$-term. The imaginary part $\operatorname{Im} Q(x-x')$ is the Fourier transform of the antisymmetric part $Q_a(k) = Q(k) - Q(-k)$ of $Q$ in momentum space. As expected, it follows from Eq. (16) that $Q_a(k)$ can be nonzero only if both $\theta$ and $\gamma_i$ are nonzero simultaneously, that is, if the spin Hall angle and the exchange field of the ferromagnet are both finite. Moreover, it should be noted that this odd in momentum term is allowed only because the top surface and bottom are different, otherwise the contributions of these surfaces to terms that odd in $k$ cancel out. This reflects that gyrotropy is a necessary condition for the appearance of anomalous currents.

## 3.1 Normal metal with negligible spin Hall angle

To illustrate the interplay between the exchange field characterized by $\gamma_i = G_i / \sigma_N$ and SOC-induced spin relaxation characterized by $l_{so}$, we first neglect the spin Hall effect, $\theta = 0$. $G_i$ describes an interfacial exchange field, and therefore we expect a critical current behavior similar to superconductor-ferromagnet-superconductor junctions, where $0$-$\pi$ transitions are possible [122]. As we show in this section, this is possible only when $\gamma_i$ is larger than a threshold value, which depends on the spin orbit relaxation and thickness of the junction.

For $\theta = \gamma_{r,s} = 0$ the kernel, Eq. (16), reduces to

$$Q(k) = d \frac{(k^2 + \kappa_t^2)S + \gamma_i^2 C}{(k^2 + \kappa_s^2)(k^2 + \kappa_t^2)d^2 S + \gamma_i^2 C}. \tag{19}$$

The poles of this expression satisfy

$$k_{10}^2 = -\kappa_s^2 - \frac{1}{2l_{so}^2} + \frac{1}{2l_{so}^2}\sqrt{1-a}, \tag{20}$$

$$k_{20}^2 = -\kappa_s^2 - \frac{1}{2l_{so}^2} - \frac{1}{2l_{so}^2}\sqrt{1-a}, \tag{21}$$

$$a = 4\frac{\gamma_i^2 l_{so}^4}{d^2} \frac{d\kappa_t}{\tanh d\kappa_t}. \tag{22}$$

The appropriate sign of $k_{10,20}$ is determined by the convergence of the integrand in the half-plane in which the contour of integration is closed, that is, determined by the condition $\operatorname{Im}(k_{10,20}(x - x')) < 0$. We identify two regimes based on whether the square root in Eq. (20) is real or imaginary, determined by $a < 1$ or $a > 1$. If the square root is imaginary, i.e. $a > 1$, the poles are complex and satisfy $k_{20} = -k_{10}^*$. The kernel in real space $Q(x - x')$ can be calculated by summing residues at $k_{1,2}$. The current follows from Eq. (10):

$$j(\omega) = \frac{1}{d}\gamma_B^2 \frac{\Delta^2}{\omega^2 + \Delta^2} \sin\phi \frac{2\pi l_{so}^2}{\sqrt{a-1}} \operatorname{Re}\left[(\kappa_t^2 + k_{10}^2 + \gamma_i^2 C/S)\frac{e^{ik_{10}L}}{k_{10}^3}\right]. \tag{23}$$

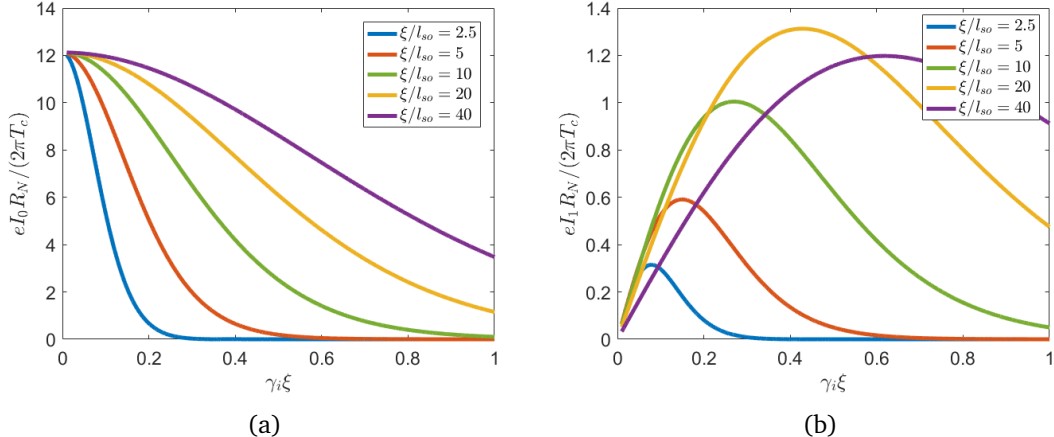

Figure 3: The $\sin\phi$ (a) and $\cos\phi$ (b) contributions of the Josephson current as a function of $\gamma_i$ for different $l_{\text{so}}$. The $\sin\phi$ contribution is suppressed by the ferromagnetic insulator, the cosine contribution is absent without the ferromagnetic insulator, but suppressed for large couplings. The maximal $I_1$ depends non-monotonically on $\xi/l_{\text{so}}$. On the other hand, $I_1$ increases with increasing $\xi/l_{\text{so}}$ for nonzero $\gamma_i$, approaching $I_0(h=0)$, which is independent of $l_{\text{so}}$. Other parameters are $\gamma_r = 0, T/T_c = 0.1, d/\xi = 0.1, L/\xi = 5$ and $\theta = 0.15$.

These expressions contain the oscillating functions $e^{ik_{10,20}L}$. Thus, the junction exhibits 0-$\pi$ transitions.

On the other hand, if $a < 1$, $k_{10,20}^2$ are both real and negative. In this limit, both poles are on the imaginary axis. Again summing the residues to obtain the kernel from Eq. (10), we obtain:

$$j(\omega) = \frac{1}{dS}\gamma_B^2 \frac{\Delta^2}{\omega^2 + \Delta^2} \sin\phi \frac{\pi l_{\text{so}}^2}{\sqrt{1-a}}\left[(\kappa_t^2 + k_{10}^2 + \gamma_i^2)\frac{e^{-|k_{10}|L}}{|k_{10}|^3} - (\kappa_t^2 + k_{20}^2 + \gamma_i^2 C/S)\frac{e^{-|k_{20}|L}}{|k_{20}|^3}\right]. \quad (24)$$

Since $|k_{10}| < |k_{20}|$, the expression between square brackets is always real and positive. Thus, there are no 0-$\pi$ transitions.

In short, 0-$\pi$ transitions appear only for large enough interfacial exchange, precisely when $a > 1$. In the absence of spin-orbit relaxation, i.e. $1/l_{\text{so}} \to 0$, even very small exchange fields lead to 0-$\pi$ transitions. In contrast, for strong spin-orbit relaxation, i.e. $l_{\text{so}} \to 0$, 0-$\pi$ transitions are completely suppressed. On the other hand, for thinner junctions smaller exchange fields are needed. A related effect of the spin-orbit coupling is that it weakens the suppression of the critical current by an exchange field [123, 124]. This can be understood from the expression for the poles in Eqs. (20-22). $\gamma_i$ only appears in 22, and this term is suppressed for small $l_{\text{so}}$.

## 3.2 Normal metal with spin Hall effect

We now focus on the case, when the combination of SOC, via the spin Hall angle $\theta \neq 0$, and magnetic proximity effect described by $G_i$, leads to an anomalous current and hence to a $\phi_0$-junction. From Eq. (16), if $\theta$ is nonzero, also terms of odd order in $k$ appear in the denominator of $Q(k)$ and hence anomalous currents may exist.

As before, we consider the case where $\kappa_s d \ll 1$, that is, singlets decay on a scale much larger than the thickness and therefore the normal metal is fully proximitized. We do no assumption on the decay length scale of the triplet correlations compared to the thickness junction, instead we explore the dependence on their ratio. An analytical compact expression

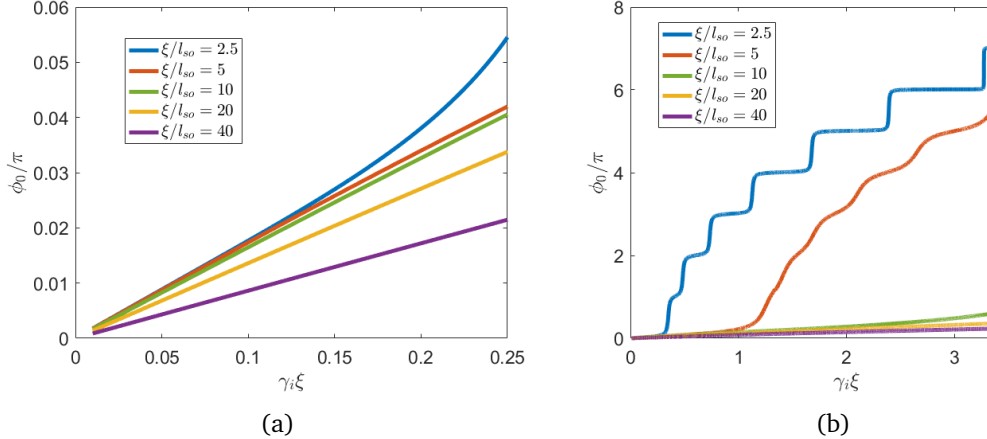

Figure 4: The anomalous phase $\phi_0$ as a function of $\gamma_i$ for different $l_{\text{so}}$. If $\gamma_i$ is small, such that $a = \frac{4\gamma_i^2 l_{\text{so}}^4}{d^2}\frac{d\kappa_t}{\tanh d\kappa_t} < 1$ and the square root in Eqs. (20,21) is real, $\phi_0$ increases gradually, as shown in panel (a). The $\phi_0$-effect decreases with stronger spin-orbit relaxation. If $a > 1$ and the square root is imaginary, there are 'smoothened' $0 - \pi$ transitions, as shown in panel (b). The stronger the spin-orbit coupling, the stronger the field at which the transition between these regimes appears. Parameters are set to $\gamma_r = 0, T/T_c = 0.1, d/\xi = 0.1, L/\xi = 5$ and $\theta = 0.15$.

for the spectral current can be obtained for enough large and small parameter $a$, Eq. 22, specifically when $|a-1| \gg \theta$. Details of the calculations and corrections for $a \approx 1$ are discussed in Appendix B.

The poles of Eq. (16) in the considered case can be written as $k_j = k_{j0} + \theta\delta k_j$, where $j = 1, 2$, and the poles at zeroth order in $\theta$, $k_j$, are defined by Eqs. (20-22), whereas

$$\delta k_1 = -\delta k_2 = \delta k = \gamma_i \frac{1 - \cosh\kappa_t d}{d\sinh\kappa_t d}\frac{l_{so}}{\sqrt{1-a}}\,, \tag{25}$$

where it is assumed that $\theta|\delta k_j| \ll k_{0j}$.

The current, calculated from Eq. (10), can be obtained by summing the residues $R_1 e^{i(k_1+\theta\delta k_1)(x-x')}$ and $R_2 e^{i(k_2+\theta\delta k_2)(x-x')}$ of $Q(k)e^{ikx}$ at the two poles after closing the integration contour in the appropriate half-plane, where $Q(k)$ is given by Eq. (16). This results in the following general expression for the spectral current:

$$j(\omega) = \gamma_B^2 \operatorname{Re} \frac{\pi\frac{\Delta^2}{\omega^2+\Delta^2}e^{i\phi}}{dS(k_1^2-k_2^2)}\left[\frac{e^{ik_1 L}}{k_1^2 k_{10}}\left\{(\kappa_t^2 + k_1^2)S + (\gamma_i^2 - 2\gamma_i\theta k_1)C\right\}\right.$$
$$\left. - \frac{e^{ik_2 L}}{k_2^2 k_{20}}\left\{(\kappa_t^2 + k_2^2)S + (\gamma_i^2 - 2\gamma_i\theta k_2)C\right\}\right]. \tag{26}$$

The total current can now be obtained by substituting Eq. (26) in Eqs. (9) and (10). Before we proceed to evaluating numerically the current, we focus on insightful analytical expressions that can be derived when $a, \theta, \kappa_s/\kappa_t \ll 1$. This case corresponds to weak charge-spin conversion, with triplets decaying over a shorter length scale than singlets. Furthermore, we consider large enough temperatures such that $k_B T \gg \Delta(T)$ and hence we may take only the first Matsubara frequency in the sums. In this case the total current has the form:

$$I(\phi) = \frac{\sigma_N}{e}\frac{2\gamma_B^2\Delta^2\xi_T^3}{dT}e^{-L/\xi_T}\left[1 + l_{so}^2\gamma_i^2\frac{d/l_{so}}{\tanh(d/l_{so})}\right]\left[\sin(\phi + \theta\delta k L) + 2\theta\delta k\xi_T\cos(\phi + \theta\delta k L)\right], \tag{27}$$

with $\xi_T = \sqrt{D/(2\pi T)}$ being the thermal length.

The anomalous phase $\phi_0$ is then given by

$$\phi_0 = -\big[\theta\delta kL + \tan^{-1}(2\theta\delta k\xi_T)\big] \approx -\theta\delta k(L+2\xi_T) = \frac{\theta\gamma_i l_{so}(L+2\xi_T)}{d}\frac{\cosh(d/l_{so})-1}{\sinh(d/l_{so})}. \quad (28)$$

Expression Eq. (28) displays the usual linear dependence on the parameter characterizing the strength of spin-orbit coupling ($\theta$) and the exchange field strength ($\gamma_i$), and length of the junction ($L$). This expression can be easily used to estimate the anomalous phase in real experiments on lateral Josephson junctions with ferromagnetic insulators, where the interfacial exchange dominates against other depairing effects ($\gamma_i \gg \gamma_r$). Moreover, we were able to incorporate spin-relaxation in our equations. For a normal metal thickness much smaller that the spin-relaxation length, *i.e.* $d/l_{so} \ll 1$, one obtains from Eq. (28) that $\phi_0 \approx \theta\gamma_i(L+2\xi_T)$. On the other hand, if $d/l_{so} \gg 1$ one obtains $\phi_0 \approx \theta\gamma_i(L+2\xi_T)l_{so}/d$. Thus, the anomalous phase, and hence the anomalous current, is suppressed by spin-orbit relaxation only if the corresponding spin relaxation length is much smaller than the thickness. This suppression appears because the anomalous current flows only in the presence of triplet correlations, that is, it is localized near the boundary with the FI on a length scale of order $l_{so}$. If the normal metal is thin enough, the triplet correlations are present in the whole junction and hence spin-orbit relaxation weakly affects the $\phi_0$-effect. Since in one-dimension the thickness does not play a role such robustness against spin-orbit relaxation is due to the inherently two-dimensional geometry.

Beyond the above limiting case the current can be calculated by numerically evaluation of (26). The results for the $\sin\phi$-contribution $I_0$ and $\cos\phi$-contribution $I_1$ to the current and $\varphi_0$ obtained from these results are shown respectively in Figs. 3 and 4 as a function of the interfacial exchange field parameter $\gamma_i$ and different values of $1/l_{so}$. A decrease of $l_{so}$ weakens the suppression of the $\sin\phi$-contribution of the current $I_0$ as shown in Fig. 3(a). Indeed, spin-orbit relaxation suppresses the effective exchange coupling of the ferromagnetic insulator and the normal metal, as discussed in the previous section. Therefore, for small $l_{so}$ the $\sin\phi$-contribution of the currents converges towards the zero-field value, as expected. The stronger the exchange field coupling, the stronger the spin-relaxation necessary to recover the current.

Fig. 3(b) shows the anomalous current $I_1$ as a function of the interfacial exchange field, $\gamma_i$. It shows a non-monotonic behaviour with a maximum at an optimum value of $\gamma_i$. Both the value of $\gamma_i$ for which $I_1$ is maximized and the maximal $I_1$ increase with decreasing $l_{so}$. Indeed, for small $\gamma_i$ the generated anomalous current is almost independent of $l_{so}^2$, and increases approximately linearly with $\gamma_i$. For larger values , $\gamma_i$ suppresses the singlet and hence there are less singlet-triplet conversion and hence the SHE is less effective. The decay of the current due to $\gamma_i$ induced pair breaking can be weaken via spin-orbit relaxation. Thus, for stronger spin-orbit relaxation the maximal anomalous current is obtained for larger values of $\gamma_i$. If $d \lesssim l_{so}$ this leads to an increase in the maximally attainable anomalous current, as shown in Fig. 3(b). However, if $l_{so} \ll d$, the maximal anomalous current is suppressed. Indeed, as explained in the analytical limit, in the presence strong spin-orbit relaxation the anomalous current only flows in a rather narrow region near the boundary with the FI.

The increase of the anomalous current with increasing spin-orbit coupling does not mean that the $\phi_0$-effect increases with decreasing $l_{so}$, as shown in Fig. 4(a). The $\phi_0$-effect is not solely determined by $I_1$, but also by $I_0$. For small exchange fields ($\gamma_i\xi \ll 1$), $a \ll 1$, the anomalous phase $\phi_0$ is almost independent of $l_{so}$, consistent with Eq. 28.

When $a$ approaches 1, the suppression of $I_0$ leads to an enhancement of the $\phi_0$-effect. Since $a \propto l_{so}^4$ smaller fields are needed for weak relaxation, that is, $\phi_0$ is larger for weak relaxation. As the exchange field is increased further the regime $a > 1$ is obtained, and the signs of $I_0$ and $I_1$ start to alternate. As discussed in Sec. 3.1 this leads to $0-\pi$ transitions in

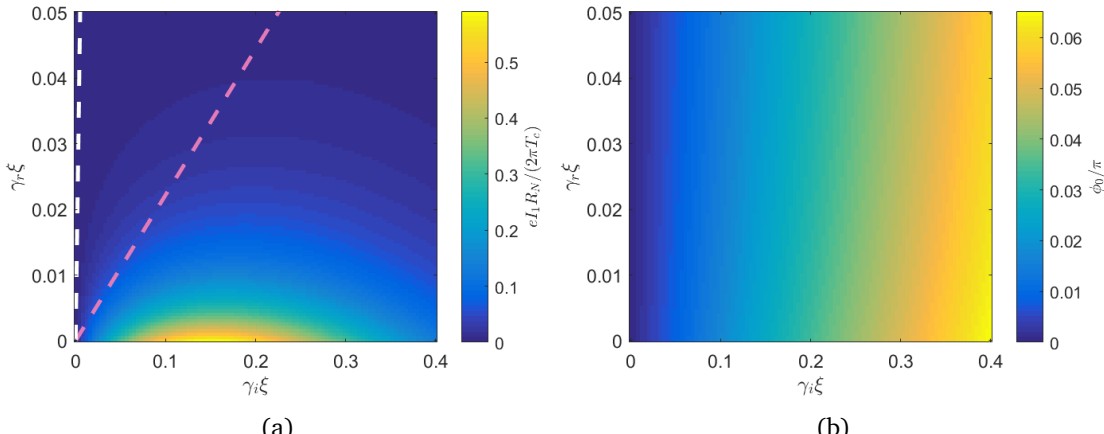

Figure 5: The dependence of the anomalous current (a) and $\phi_0$ (b) in the Josephson junction on the boundary parameters $\gamma_r$ and $\gamma_i$. While the anomalous current falls off rapidly as a function of $\gamma_r$ and is maximized at a finite $\gamma_i$, the $\phi_0$-parameter increases with $\gamma_i$ and decreases slowly with $G_r$. The pink dashed line is used to indicate $G_i/G_r \approx 4.5$, as is the case for EuS, while the white dashed line is used to indicate $G_i/G_r \approx 0.1$, as is the case for YiG. Using suitable superconductors and normal metals one may move over these lines in the diagram. In the case of YiG no significant anomalous current can be obtained. Using EuS the attainable anomalous current is an order of magnitude larger. Other parameters are $\gamma_r = 0, T/T_c = 0.1, d/\xi = 0.2, L/\xi = 5$ and $\theta = 0.15$.

the case $\theta = 0$, i.e. sharp transitions between $\phi_0 = 0$ and $\phi_0 = \pi$. Fig. 4(b) shows that for finite $\theta$ these transitions are smoothened, and $\phi_0$ takes intermediate values. One should bear in mind, however, that in the regime of large $\gamma_i$ shown in Fig. 4(b) both the values of $I_0$ and $I_1$ are very small.

## 3.3 Interfacial pair-breaking: The effect of $G_r$

In the previous section, we focused on ferromagnetic insulators with large interfacial exchange field parameter $\gamma_i \gg \gamma_{r,s}$. In other words, we neglected the terms $G_r$ and $G_s$ in the boundary condition, Eq. (8). As explained above, these terms take the form of a pair-breaking term, similar to magnetic impurities. and hence, they suppress not only the triplet, but also the singlet component induced by the proximity effect in N.

The prototypical magnetic insulator used in spintronics is yttrium iron garnet (YIG), a ferrimagnet with compensated magnetic moment for which $G_r > G_i$, and hence $\gamma_r > \gamma_i$. In this section, we quantify the effect of $G_r$ (at low temepratures one can neglect the effect of $G_s$ [107].)

The kernel $Q(k)$, given in Eq. (16), is still a rational function, and the same procedure as in previous section can be applied to find the poles and currents to first order in $\theta$. In the limit $a, \theta, \kappa_s/\kappa_t \ll 1$, $\phi_0$ has the same form as in Eq. 28, but with a renormalized thermal length $\xi_T^{-2} \to \xi_T^{-2} + \gamma_r/d$ and spin orbit relaxation length $l_{so}^{-2} \to l_{so}^{-2} + \gamma_r/d(\kappa_t d/\tanh d\kappa_t - 1)$. Thus, as expected, the effect of $\gamma_r$ is to reduce both the proximity penetration length and the effective spin-relaxation length, thereby suppressing the $\phi_0$-effect.

The dependence of the anomalous current and $\phi_0$ on $\gamma_{i,r}$ is illustrated in Fig. 5. As before, a finite $\gamma_i$ is needed to have an anomalous current in the system, but a large $\gamma_i$ suppresses this anomalous current since the proximity effect becomes weak, see Fig. 5(a), In addition, the pair breaking term $\gamma_r$ suppresses the anomalous current regardless of $\gamma_i$. For $\gamma_i \ll \gamma_r$

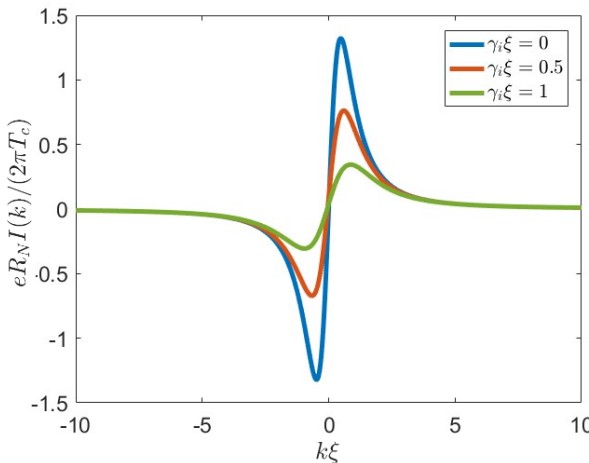

Figure 6: The current $I(k)$ flowing through the N layer in the trilayer junction when the pair potential in the superconductor is given by $\Delta = \Delta_0 e^{ikx}$ is shown for different $G_i$. There is a maximum of the induced current that is attained for a finite phase gradient. If the current through the supercurrent and hence $k$ are increased further the current in the normal metal decreases towards 0. Other parameters are $\gamma_r = 0, T/T_c = 0.1, d/\xi = 0.2, L/\xi = 5$ and $\theta = 0.15$.

the anomalous current almost vanishes. The ratio between $\gamma_i$ and $\gamma_r$ for two used materials, the ferromagnetic insulator EuS and the ferrimagnet YIG are indicated using dashed lines in the 2D plot. While for YIG the obtained anomalous current is almost negligible, for EuS sizeable anomalous currents can be obtained. Since $\xi$ is determined by the superconductor and the normal metal, by choosing different materials one may obtain different points on this line. For typical values [107], $G_{i,r} \sim 10^{13}\ \Omega^{-1}\text{m}^{-2}$, $\sigma_N \sim 10^8\ \Omega^{-1}\text{m}^{-1}$, the density of states $N_0 \sim 10^{28}/10^{-18} = 10^{46}\ \text{m}^{-3}\text{J}^{-1}$ and $\Delta \sim 1$ meV ($T_c \sim 5$ K), one obtains $\gamma_{i,r}\xi \sim 0.1$, which is within the optimal range. The effect on $\phi_0$ is illustrated in Fig. 5(b). $\phi_0$ increases monotonically with $\gamma_i$. This increase is suppressed by $\gamma_r$, but the suppression is weaker than for the anomalous current $I_1$.

# 4 The diode effect

As mentioned when writing Eq. (17), the linearization procedure leads to a current phase relation which only contains the first harmonics in $\phi$. This prevents obtaining the diode effect [65]. To describe the diode effect in the Josephson junction of Fig. 2 one either needs to solve the full non-linear boundary problem numerically, Eqs. (5-8), or to expand the solution up to next leading order in the proximity effect. These tasks are beyond the scope of the present work. Instead we address the diode effect in a slightly different structure for which it is enough to use the linearized equations to calculate non-reciprocal transport effects. The geometry is shown in 2(b). It is closely related to the setup described in the previous section. Just like the Josephson junction, it consist of a S/N/FI trilayer. Thus, also this structure is gyrotropic, it has the z-axis as its polar vector. In fact, since the previous problem was solved in Fourier space, the equations describing this structure are the same as used in previous sections. This means that we may use the same kernel $Q(k)$ as presented in Eq. (19), but now use the same kernel to describe the system. What differs is the quantity that we calculate from this kernel.

Indeed, contrary to the Josephson junction, the superconductor now fully covers the normal metal. If a current passes through the superconductor, a phase gradient $k$ develops, that is, the pair potential in the superconductor has the form $\Delta = \Delta_0 e^{ikx}$. In this case, the current through the normal metal depends of the phase gradient, $I(k)$. As we show next, in the presence of SOC, the maximum value of that current depends on its direction. This explains a kind of diode effect which can be described even in the linearized case.

In our calculation, we assume that the current passing through the superconductor is much smaller than its critical current, so that we do not need to determine the gap self-consistently, that is, $\Delta_0$ is independent of $k$ for the range of $k$ considered. Under this assumption the equations to be solved for the anomalous Green's function are exactly those used to find the kernel of the Josephson junction in Sec. 3.2. Indeed, the corresponding boundary value problem for this setup is exactly the one given by Eqs. (14-15). The solution to this boundary value problem is presented on in appendix Sec. A and given by Eqs. (A.10-A.13) therein. Here we show the results for the current using Eq. (6). As in the previous sections, $\check{g}$ commutes with $\boldsymbol{m} \cdot \sigma$ and hence the spin-swapping term does not contribute. Hence, there are two contributions to the current $D\partial_x \check{g}$ and $2D\theta \partial_z \check{g}\sigma_y$. Going to the linearized limit and converting to momentum space in the $x$-direction, Eq. (6) reduces to

$$\frac{eR_N}{2\pi T_c} I(k) = \frac{T}{T_c} \sum_n \int_{-d}^{0} k \operatorname{Re}(F_s^* F_s - F_t^* F_t) + 2\theta \operatorname{Re}(F_s^* \partial_z F_t + F_t^* \partial_z F_s)dz \,, \qquad (29)$$

where the summation is over Matsubara frequencies. The dependence of the current on the phase gradient is shown in Fig. 6. The second term represents anomalous currents, that is, currents at $k = 0$. The necessary conditions for this term to be nonzero at $k = 0$ are the presence of a spin-Hall angle $\theta$, the generation of triplet correlations, possible via the exchange field of the FI, and a nonzero average gradient of the pair amplitudes. The latter is allowed due to the specific gyrotropy of the junction.

Even though the current in the superconductor is unbounded due to the absence of self-consistency, there is a critical current in the normal metal. It appears because in the presence of a phase gradient $k$ the coherence length $\sqrt{D/|\omega|}$ is modified to $\sqrt{D/\sqrt{\omega^2 + k^2}}$, that is, the proximity effect is weaker if the phase gradient is larger. This limits the current flowing trough N for large $k$. That is, the Cooper pairs in the superconductor can "drag" only a finite supercurrent in the normal layer. If either $\theta = 0$ or $\gamma_i = 0$, this maximum Cooper drag is the same in both directions. However a finite $\theta$ and $\gamma_i$ the maxima and minima at positive and negative $k$ respectively are not necessarily the same. In the parameter regime discussed here, this effect is small but finite, see Fig. 6.

To have a qualitative comparison of the homogeneous junction in Fig. 2(b) with the lateral Josephson junction in Fig. 2(a), the anomalous current is calculated as $I(k = 0)$. As shown in Fig. 7, the anomalous current is non-monotonous as a function of $\gamma_i$ and suppressed by increasing $\gamma_r$, similar to the anomalous current in the Josephson junction. The anomalous current is maximized for $\gamma_i \xi \approx 0.5$, which is in good correspondence with the value of $\gamma_i$ for which the anomalous current is maximized in the Josephson junction in Fig. 3(b).

The quality of the diode effect is characterized by the diode efficiency $\eta$, defined as

$$\eta = \frac{\max_k I(k) - |\min_k I(k)|}{\max_k I(k) + |\min_k I(k)|} \,. \qquad (30)$$

The dependence of the diode efficiency on exchange field strength is illustrated for different thicknesses in Fig. 8(a). As for the anomalous current discussed in Sec. 3.2, the thinner junction the smaller the exchange field for which the diode efficiency is maximized. For large exchange fields the diode efficiency may even change sign. However, for too large $\gamma_i$ the

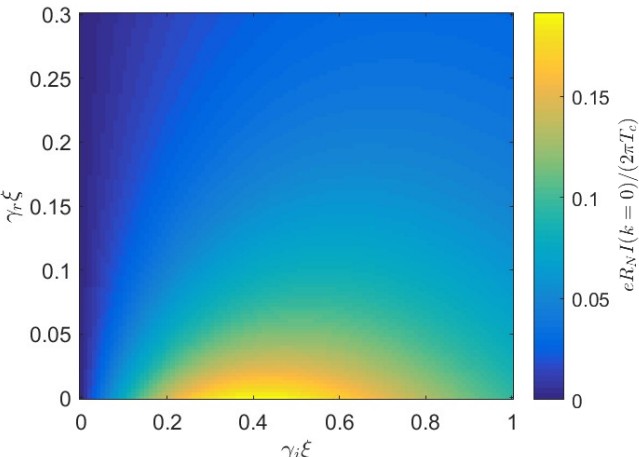

Figure 7: The anomalous current $I(k = 0)$ in the trilayer junction as a function of $\gamma_r$ and $\gamma_i$. Other parameters are $\xi/l_{so} = 5, T/T_c = 0.1, d/\xi = 0.2, L/\xi = 5$ and $\theta = 0.15$.

critical current is highly suppressed, and hence this parameter regime may be less suitable for experiment.

The dependence of $\eta$ on $l_{so}$ is shown for $\gamma_i\xi = 0.5$ in Fig. 8(b). Similar to the results for the anomalous current discussed in the previous section (Fig. 3(b)), decreasing $l_{so}$ leads to an increase in the maximal currents, which is beneficial for the diode efficiency. The diode efficiency however, does not increase indefinitely with decreasing $l_{so}$, see Fig. 8(b). Indeed, if $l_{so}$ is much smaller than the thickness, triplet correlations exist only in a small region near the interface and hence the non-reciprocal part of the current is supported only in part of the junction. This leads to a non-monotonous dependence of the diode efficiency on $l_{so}$.

Finally, for interfaces with a finite pair-breaking parameter, $\gamma_r$, we show in Fig. 9 the maximum critical current and diode efficiency as functions of $\gamma_{r,i}$. Both effects depend non-monotonically on $\gamma_i$. They vanish for $\gamma_i = 0$ but increase with increasing $\gamma_i$, reaching a maximum, and then decrease with further increases in $\gamma_i$. In contrast, $\gamma_r$ suppresses the critical current monotonically as it weakens the singlet correlations, as seen in Fig. 9a. The suppression of $\eta$ by $\gamma_r$ is, however, weaker, as shown in Fig. 9b.

The values obtained here for $\eta$ serve as lower bounds, given that we have linearized the equations under the assumption of a weak proximity effect. Consequently, in experiments with good SC/N interfaces and a strong proximity effect, one can anticipate observing larger values for $\eta$. For two materials EuS and YiG, typical ratios $G_i/G_r$ have been indicated using dashed lines in 9. For EuS (pink), significant diode effects can be obtained by choosing suitable combinations of superconductor and normal metal, for YiG (white) the diode efficiency is typically small.

## 5   Discussion and conclusions

According to our model, optimizing the anomalous current, and therefore the diode effect, requires a normal metal with significant spin-orbit coupling and an interfacial exchange field (described by the parameter $G_i$) between the N and the FI in such a way that it dominates over pair-breaking effects related to spin relaxation at this interface. Heavy metals like Pt and Ta have shown to possess a large spin Hall angle and a good charge-spin interconversion

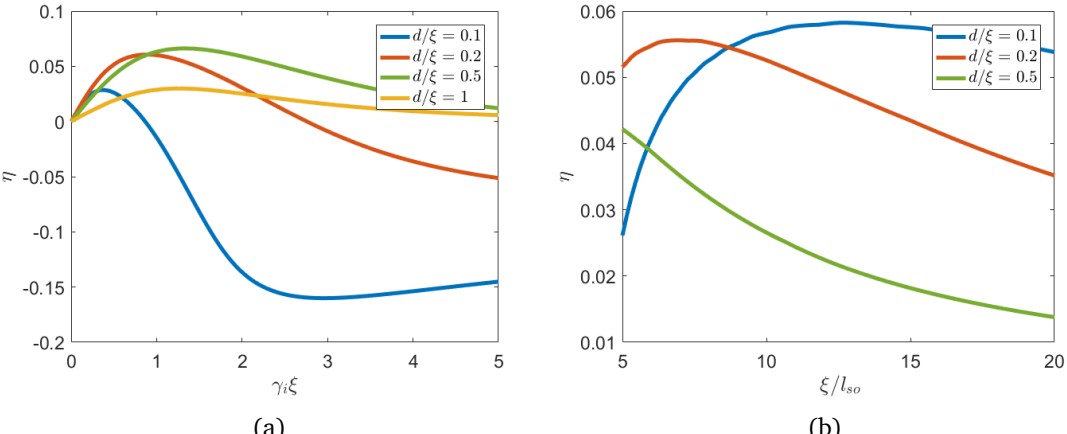

Figure 8: (a): The diode efficiency $\eta$ in the trilayer junction in positive as a function of $\gamma_i$ for different thicknesses. The diode efficiency may change sign as a function of $\gamma_i$ for thin enough junctions. Other parameters are set to $\gamma_r = 0$, $T/T_c = 0.1$, $\xi/l_{so} = 5$, $L/\xi = 5$ and $\theta = 0.15$. (b): The diode efficiency as a function of $\xi/l_{so}$ for different thicknesses of the junction. The efficiency has an optimum for finite spin-orbit relaxation, when the spin orbit relaxation length is of the same order as the thickness of the material. For smaller relaxation the exchange field suppresses the current, for larger relaxation the region in which the anomalous current may flow is constricted. Other parameters are $\gamma_r = 0$, $T/T_c = 0.1$, $\gamma_i \xi = 0.5$, $L/\xi = 5$ and $\theta = 0.15$.

and are therefore suitable to serve as the normal metal in such junctions. As an FI material, the well-studied ferromagnetic insulator EuS [125] is recommended. For example, Ref. [121] reported SMR measurements in Pt/EuS systems and established high values for $G_i$. Hence, it is anticipated that a Josephson junction like the one depicted in Fig. 2(a) with this material combination could exhibit very significant magnetoelectric effects.

In Ref. [100], nonreciprocal effects were reported in Josephson junctions with the same geometry studied in the present work. Instead of using a FI, they employed YIG, a well-known ferrimagnet often studied in combination with Pt [108, 126–130]. SMR measurements combined with the existing theory establish that for this material combination, $G_r \gg G_i$. According to our theory, in this case, both anomalous current and diode effect are strongly suppressed. Nevertheless, the experiment in Ref. [100] suggests a diode effect with notable efficiency. Some explanations for this discrepancy can be considered. Firstly, it is well-known that the values of spin-mixing conductance critically depend on the Pt/YIG interface preparation. It is possible that in the experiments from Ref. [100], this interface significantly differs from those in experiments showing a clear SMR signal. On the other hand, the origin of terms proportional to $G_r$ in Eq. (8) could be due to strong spin-orbit coupling at the interface, as postulated in Ref. [100]. This type of relaxation does not affect the singlet component of the condensate, and thus, the diode effect would persist despite a large spin relaxation. Thus, in the superconducting state one can distinguish the origin of spin-relaxation by studying nonreciprocal transport and anomalous currents. In addition, strong interfacial SOC may lead to an enhancement of the effective spin-Hall angle $\theta$ near the interface and hence to a larger SHE and anomalous current.

To elucidate these points and establish a comprehensive understanding of such junctions, we propose an experiment that measures, in the same system, both the angle-dependent magnetoresistance, and the nonreciprocal superconducting effect. For the first experiment, mea-

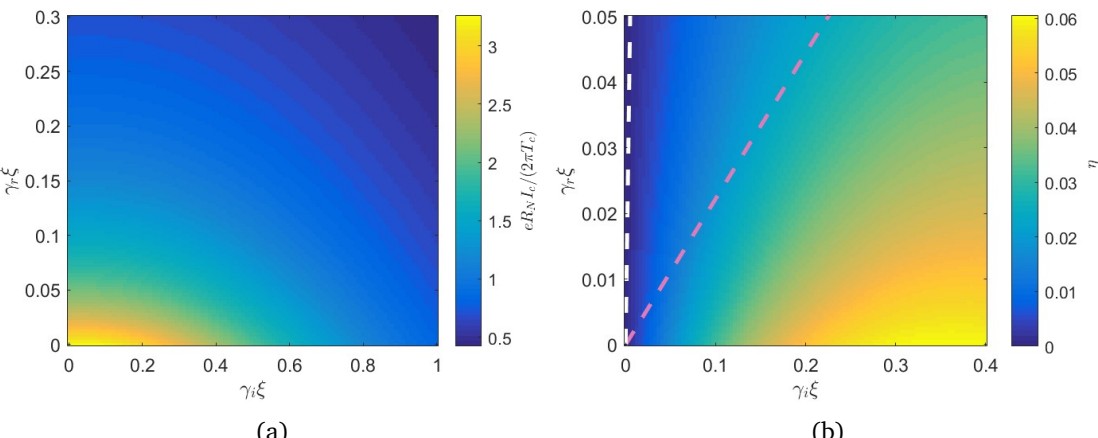

Figure 9: (a): The maximum critical current $I_c = \max(I_{c+}, I_{c-})$ in the trilayer junction is suppressed as a function of $\gamma_{r,i}$. The suppression by $\gamma_r$ is significantly stronger than the suppression by $\gamma_i$. (b): The diode efficiency as a function of $\gamma_r$ and $\gamma_i$. A nonzero $\gamma_i$ is needed to have a diode effect, $\gamma_r$ suppresses the diode efficiency. Other parameters are $\xi/l_{so}^2 = 5, T/T_c = 0.1, d/\xi = 0.2, L/\xi = 5$ and $\theta = 0.15$. The pink dashed line is used to indicate $G_i/G_r \approx 4.5$, as is the case for EuS, while the white dashed line is used to indicate $G_i/G_r \approx 0.1$, as is the case for YiG. Using suitable superconductors and normal metals one may move over these lines in the diagram.

surements should be performed above the superconducting critical temperature to avoid effects related to the superconducting proximity effect. The interfacial coefficients $G_{i,r,s}$ may be determined using the SMR theory [99]. One can then measure the non-reciprocal transport when $T < T_c$ and try to use the above parameters to fit the results. This experiment would shed light on the connection between magnetoelectric effects in normal and superconducting systems, which so far have only been studied separately. This would provide an opportunity to a better understanding of the primary microscopic mechanisms at N/FI interfaces and an interesting testbed for existing theories.

In summary, we have presented an exhaustive study of the simplest structure with a polar symmetry exhibiting magnetoelectric and nonreciprocal transport effects in the absence of an external applied magnetic field. The suggested structures are lateral N/FI bilayers either embedded in a Josephson junction or forming a trilayer with a superconductor. The gyrotropy in our system is not a microscopic property. In fact, all the materials we have considered exhibit inversion symmetry. However, gyrotropy arises due to the presence of the hybrid interface. In other words, gyrotropy does not necessarily have to be a microscopic characteristic of the system; it can be induced at the mesoscopic or even macroscopic scale.

The combination of the induced superconductivity via the proximity effect, the SOC in the N layer and the interfacial exchange at the N/FI leads to an anomalous supercurrent that manifest as anomalous phase in the Josephson configuration and as the diode effect. Our theoretical framework based on an effective action of the non-linear $\sigma$-model of the system, allowed us to make connection between the interfacial spin-mixing conductance formalism and the superconducting proximity effects in mesoscopic structures.

In particular, we explore the effect of the spin-relaxation which is unavoidable in metals like Pt or Ta with strong SOC. Both the anomalous current and the $\phi_0$-effect survives even in the presence of strong spin orbit relaxation, as long as the thickness of the N layer is comparable to the spin relaxation length. However, there is no restriction on the length of the junction: large anomalous currents and diode effects are obtained even if the length of the junction is much

larger than the spin relaxation length. This provides a major step forward in the engineering of Josephson diodes compared to one-dimensional SDEs with Rashba-like spin-orbit coupling. We also demonstrated how these effects depend on the spin conductance parameters describing the N/FI interface, finding that the diode effect is maximized at interfaces with large exchange coupling, $G_i > G_r$.

We have also calculated non-reciprocal transport in a different setup, which exploits the maximum Cooper drag of a superconductor on a metal proximized by this superconductor. We have shown that this problem has the same kernel as the Josephson junction, and therefore the anomalous current in this setup is similar to that in the Josephson junction. Moreover, in this geometry the diode effect can be studied analytically using linearized equations, unlike in other types of junctions. The correspondence with the $\phi_0$-effect in junctions allows us to use the results obtained from linearized equations as proof of diode effects in lateral Josephson junctions with an FI.

## Acknowledgments

We would like to thank Felix Casanova, V.N. Golovach and A.A. Golubov for useful discussions.

**Funding information** T.K. and F.S.B. acknowledge financial support from Spanish MCIN/AEI/ 10.13039/501100011033 through project PID2020-114252GB-I00 (SPIRIT) and TED2021-130292B-C42, and the Basque Government through grant IT-1591-22. I.V.T. acknowledges support by Grupos Consolidados UPV/EHU del Gobierno Vasco (Grant IT1453-22) and by the grant PID2020-112811GB-I00 funded by MCIN/AEI/10.13039/501100011033.

## A    Linearized equation and kernel

In this appendix we derive the linearized Usadel equation for the lateral system in Fig. 2 and present the equations from which the kernel used in the main text may be calculated. We give explicit expressions for the solutions of those equations, thereby showing how the kernel is obtained.

In the limit $\gamma_B/\kappa_s \ll 1$ the Usadel equation may be expanded to lowest order in the pair amplitudes. That is, the Green's function can be approximated as

$$G \approx \begin{bmatrix} \text{sign}(\omega) & F_s + iF_t\text{sign}(\omega)\sigma_y \\ F_s^* - iF_t^*\text{sign}(\omega)\sigma_y & -\text{sign}(\omega) \end{bmatrix}. \tag{A.1}$$

The triplet-terms $F_t$ are multiplied by $\text{sign}(\omega)$ because the triplet correlations are odd-frequency in contrast to the even frequency singlet correlations. Indeed, from Eqs. 5 and 7 it follows that the Green's function satisfies the symmetry $\check{g}(-\omega) = -\tau_3\sigma_y\check{g}(\omega)^*\sigma_y\tau_3$. By taking out the $\text{sign}(\omega)$, $F_s$ and $F_t$ thus satisfy the same symmetry with respect to $\omega$, $F_{s,t}(-\omega) = F_{s,t}(\omega)^*$. By substituting this parameterization in the Usadel equation and the boundary conditions, given by Eqs. 5,6 and 7 in the main body, the following linearized equations for $F_s, F_t$ are obtained:

$$\nabla^2 F_s = 2\omega/D F_s, \tag{A.2}$$

$$\nabla^2 F_t = \left(2\omega/D + \frac{1}{l_{so}^2}\right)F_t, \tag{A.3}$$

with boundary conditions

$$
\begin{aligned}
& -\partial_z F_s(z=0) + \theta \partial_x F_t(z=0) = D\gamma_B f_{s0}, \\
& -\partial_t F_s(z=0) - \theta \partial_x F_s(z=0) = 0, \\
& -\partial_z F_s(z=-d) + \theta \partial_x F_t(z=-d) = 2D\gamma_i F_t(z=-d) + 2D\gamma_r F_s(z=-d), \\
& -\partial_t F_s(z=-d) - \theta \partial_x F_s(z=-d) = 2D\gamma_i F_s(z=-d) + 2D\gamma_r F_t(z=-d).
\end{aligned}
\tag{A.4}
$$

For the linearized Usadel equation, the current that passes through the superconductor-normal metal interface at a given coordinate $x$ may be expressed via the Kuprianov-Luckichev boundary conditions as

$$
j(\omega, x) = \gamma_B \mathrm{Tr}(\tau_3[\check{g}_s, \check{g}]) = 4\gamma_B \mathrm{Im}(f_{s0}(x)^* F_s(x)),
\tag{A.5}
$$

where $f_{s0}(x)$ is the pair amplitude in the superconductor at point $x$ and $F_s(x)$ is the pair amplitude in the normal metal at the interface at position $x$. The kernel is defined in momentum space via the relation $F_s(k) = \gamma_B Q(k) f_{s0}(k)$. Using the convolution rules for Fourier transforms, the corresponding expression in real space is

$$
F_s(x) = \int_{-\infty}^{\infty} F_s(k) e^{ikx} dk = \gamma_B \int_{-\infty}^{\infty} Q(k) f_{s0}(k) e^{ikx} dk = \gamma_B \int_{-\infty}^{\infty} Q(x-x') f_{s0}(x') dx'. \tag{A.6}
$$

Substituting Eq. (A.6) into Eq. (A.5), it follows that

$$
j(\omega, x) = \frac{1}{4} \mathrm{Tr}\tau_3(\check{g}\nabla\check{g}) = \gamma_B^2 \mathrm{Im} \int_{-\infty}^{\infty} f_{s0}(x)^* Q(x-x') f_{s0}(x') dx'. \tag{A.7}
$$

Since $f_{s0}(x) = f_L \Theta(-x - L/2) + f_R \Theta(x - L/2)$, the total current flowing between the right superconducting electrode and the normal metal can be expressed as

$$
\begin{aligned}
j(\omega) &= \int_R j(\omega, x) dx = 4\gamma_B^2 \mathrm{Im} \int_{-\infty}^{\infty} \int_R f_{s0}(x)^* Q(x-x') f_{s0}(x') dx' dx \\
&= 4\gamma_B^2 \mathrm{Im} \int_L \int_R f_L^*(\omega) Q(x-x') f_R dx' dx + 4\gamma_B^2 \mathrm{Im} \int_R \int_R f_R^* Q(x-x') f_R(\omega) dx' dx, \tag{A.8}
\end{aligned}
$$

where the notation $\int_L$ and $\int_R$ is used to denote integration over the left and right electrode respectively. The second term should be ignored because it does not represent currents that flow from left to right, but currents both leaving and re-entering the right electrode. This results in the following expression for the current:

$$
j(\omega) = \gamma_B^2 \mathrm{Im}\left\{ \int_L dx \int_R dx' f_L^* Q(x-x') f_R \right\},
$$

which is presented as Eq. (10) in the main body. The Fourier component $Q(k)$ of the kernel is calculated from the solution at $z = 0$ of the following boundary value problem,

$$
\begin{aligned}
& \partial_z^2 F_s(\omega, k, z) - (k^2 + \kappa_s^2) F_s(\omega, k, z) = 0, \\
& \partial_z^2 F_t(\omega, k, z) - (k^2 + \kappa_t^2) F_t(\omega, k, z) = 0, \\
& \partial_z F_s(\omega, k, 0) + k\theta F_t(\omega, k, 0) = \gamma f_{s0}(k), \\
& \partial_z F_t(\omega, k, 0) - k\theta F_s(\omega, k, 0) = 0, \\
& \partial_z F_s(\omega, k, -d) - (\gamma_i - k\theta) F_t(\omega, k, -d) - (\gamma_r + 3\gamma_s) F_s(\omega, k, -d) = 0, \\
& \partial_z F_t(\omega, k, -d) + (\gamma_i - k\theta) F_s(\omega, k, -d) - (\gamma_r + \gamma_s) F_t(\omega, k, -d) = 0,
\end{aligned}
\tag{A.9}
$$

where $\kappa_t^2 = 2|\omega|/D + \frac{1}{l_{so}}$, $\kappa_s = \sqrt{2|\omega|/D}$. The solution to this boundary problem is given by

$$F_s(\omega, k, z) = A \cosh \sqrt{k^2 + \kappa_s^2} z + B \sinh \sqrt{k^2 + \kappa_s^2} z \,,$$
$$F_t(\omega, k, z) = C \cosh \sqrt{k^2 + \kappa_t^2} z + D \sinh \sqrt{k^2 + \kappa_t^2} z \,. \tag{A.10}$$

The coefficients $A, B, C, D$ are to be determined by the boundary conditions. The former two boundary conditions imply

$$\sqrt{k^2 + \kappa_s^2} B = \gamma_B f_{s0} - k\theta C \,, \qquad \kappa_t D = k\theta A \,. \tag{A.11}$$

Substituting this into the other two boundary conditions one finds up to first order in $\theta$ that $A$ and $C$ should satisfy the following relations:

$$\left( \sqrt{k^2 + \kappa_s^2} \sinh \sqrt{k^2 + \kappa_s^2} d + (\gamma_r + 3\gamma_s) \cosh \sqrt{k^2 + \kappa_s^2} d - \frac{\gamma_i}{\sqrt{k^2 + \kappa_t^2}} k\theta \sinh \sqrt{k^2 + \kappa_t^2} d \right) A$$
$$+ \left( \gamma_i \cosh \sqrt{k^2 + \kappa_t^2} d + k\theta \cosh \sqrt{k^2 + \kappa_s^2} d + \frac{\gamma_r}{\sqrt{k^2 + \kappa_s^2}} k\theta \sinh \sqrt{k^2 + \kappa_s^2} d \right) C$$
$$= \gamma_B f_{s0} \left( \cosh \sqrt{k^2 + \kappa_s^2} d + \frac{(\gamma_r + 3\gamma_s)}{\sqrt{k^2 + \kappa_s^2}} \sinh \sqrt{k^2 + \kappa_s^2} d \right),$$
$$\left( -\sqrt{k^2 + \kappa_t^2} \sinh \sqrt{k^2 + \kappa_t^2} d - (\gamma_r + \gamma_s) \cosh \sqrt{k^2 + \kappa_t^2} d + \frac{\gamma_i}{\kappa_s} k\theta \sinh \sqrt{k^2 + \kappa_s^2} d \right) C$$
$$+ \left( \gamma_i \cosh \sqrt{k^2 + \kappa_s^2} d + k\theta \cosh \sqrt{k^2 + \kappa_t^2} d + \frac{(\gamma_r + \gamma_s)}{\sqrt{k^2 + \kappa_t^2}} k\theta \sinh \sqrt{k^2 + \kappa_t^2} d \right) A$$
$$= \gamma_B \frac{\gamma_i}{\sqrt{k^2 + \kappa_s^2}} f_{s0} \sinh \sqrt{k^2 + \kappa_s^2} d \,. \tag{A.12}$$

Since we are interested in the current to first order in $\theta$, in the resulting expression we keep terms in numerator and denominator up to first order in $\theta$:

$$\frac{A}{\gamma_B f_{s0}}$$
$$= \frac{1}{\left( \kappa_{sk} \sinh \kappa_{sk} d + (\gamma_r + 3\gamma_s) \cosh \kappa_{sk} d \right)\left( \kappa_{tk} \sinh \kappa_{tk} d + (\gamma_r + \gamma_s) \cosh \kappa_{tk} d \right) + \gamma_i^2 \cosh \kappa_{sk} d \cosh \kappa_{tk} d + 2k\theta\gamma_h}$$
$$\times \left( \kappa_{tk} \sinh \kappa_{tk} d \cosh \kappa_{sk} d + \frac{\gamma_i^2}{\kappa_s} \cosh \kappa_{tk} d \sinh \kappa_{sk} d + (\gamma_r + 3\gamma_s) \cosh \kappa_{tk} d \cosh \kappa_{sk} d \right.$$
$$\left. + \frac{(\gamma_r + 3\gamma_s)(\gamma_r + \gamma_s)}{\kappa_{sk}} \cosh \kappa_{tk} d \sinh \kappa_{sk} d + \frac{(\gamma_r + \gamma_s)\kappa_{tk}}{\kappa_{sk}} \sinh \kappa_{tk} d \sinh \kappa_{sk} d \right),$$
$$\frac{C}{\gamma_B f_{s0}}$$
$$= \frac{1}{\left( \kappa_{sk} \sinh \kappa_{sk} d + (\gamma_r + 3\gamma_s) \cosh \kappa_{sk} d \right)\left( \kappa_{tk} \sinh \kappa_{tk} d + (\gamma_r + \gamma_s) \cosh \kappa_{tk} d \right) + \gamma_i^2 \cosh \kappa_{sk} d \cosh \kappa_{tk} d + 2k\theta\gamma_h}$$
$$\times \left( \gamma_i + k\theta \cosh \kappa_{sk} d \cosh \kappa_{tk} d + \frac{\gamma_i^2}{\kappa_{sk}\kappa_{tk}} k\theta \sinh \kappa_{sk} d \sinh \kappa_{tk} d + \frac{(\gamma_r + 3\gamma_s)k\theta}{\kappa_{sk}} \cosh \kappa_{tk} d \sinh \kappa_{sk} d \right.$$
$$\left. + \frac{(\gamma_r + \gamma_s)k\theta}{\kappa_{tk}} \sinh \kappa_{tk} d \cosh \kappa_{sk} d + \frac{(\gamma_r + 3\gamma_s)(\gamma_r + \gamma_s)k\theta}{\kappa_{sk}\kappa_{tk}} \sinh \kappa_{tk} d \sinh \kappa_{sk} d \right), \tag{A.13}$$

where $\kappa_{s,tk} = \sqrt{k^2 + \kappa_{s,t}^2}$ were introduced for brevity of notation Now recall that the kernel $Q(k)$ of the problem is defined such that

$$\gamma_B f_{s0} Q(k) = F_{s0}(\omega, k, z = 0) = A, \tag{A.14}$$

and thus we conclude that

$$Q(k)$$

$$= \frac{1}{\left(\kappa_{sk}\sinh\kappa_{sk}d + (\gamma_r + 3\gamma_s)\cosh\kappa_{sk}d\right)\left(\kappa_{tk}\sinh\kappa_{tk}d + (\gamma_r + \gamma_s)\cosh\kappa_{tk}d\right) + \gamma_i^2\cosh\kappa_{sk}d\cosh\kappa_{tk}d + 2k\theta\gamma_h}$$

$$\times \Big(\kappa_{tk}\sinh\kappa_{tk}d\cosh\kappa_{sk}d + \frac{\gamma_i^2}{\kappa_s}\cosh\kappa_{tk}d\sinh\kappa_{sk}d + (\gamma_r + 3\gamma_s)\cosh\kappa_{tk}d\cosh\kappa_{sk}d$$

$$+ \frac{(\gamma_r + 3\gamma_s)(\gamma_r + \gamma_s)}{\kappa_{sk}}\cosh\kappa_{tk}d\sinh\kappa_{sk}d + \frac{(\gamma_r + \gamma_s)\kappa_{tk}}{\kappa_{sk}}\sinh\kappa_{tk}d\sinh\kappa_{sk}d\Big). \qquad \text{(A.15)}$$

In the limit $\kappa_s d \ll 1$ this reduces in first order in $\theta$ to Eq. (16) in the main body. The poles of this expression to zeroth order in $\theta$ are given by

$$k_1^2 = -\left(\kappa_s^2 + \frac{\gamma_r}{d}\right) - \frac{1}{2}\left(1/l_{so}^2 + \frac{\gamma_r}{d}\left(\frac{d\kappa_t}{\tanh d\kappa_t} - 1\right)\right) + \frac{1/l_{so}^2 + \frac{\gamma_r}{d}\left(\frac{d\kappa_t}{\tanh d\kappa_t} - 1\right)}{2}\sqrt{1-c}\,,$$

$$\text{(A.16)}$$

$$k_2^2 = -\left(\kappa_s^2 + \frac{\gamma_r}{d}\right) - \frac{1}{2}\left(1/l_{so}^2 + \frac{\gamma_r}{d}\left(\frac{d\kappa_t}{\tanh d\kappa_t} - 1\right)\right) - \frac{1/l_{so}^2 + \frac{\gamma_r}{d}\left(\frac{d\kappa_t}{\tanh d\kappa_t} - 1\right)}{2}\sqrt{1-c}\,,$$

$$\text{(A.17)}$$

$$c = 4\frac{\gamma_i^2}{\left(1/l_{so}^2 + \frac{\gamma_r}{d}\left(\frac{d\kappa_t}{\tanh d\kappa_t} - 1\right)\right)^2 d^2}\frac{d\kappa_t}{\tanh d\kappa_t}\,. \qquad \text{(A.18)}$$

# B  Calculation of poles of the kernel

In this appendix we discuss the poles of the kernel $Q(k)$ in the presence of a finite spin Hall angle $\theta$. First we derive Eqs. 20 to 25 in the main body under the assumption that $a \ll 1$, that is, all poles are well separated, so that the correction to the poles is much smaller than the difference between the poles. Next to this, we derive the expressions for the poles in the limit where $|k_{10} - k_{20}| \ll \theta|\delta k_{1,2}|$. We will show here the case $\gamma_{r,s} = 0$. In the presence of finite boundary spin relaxation the derivation is very similar, with the result shown in the previous section. The final expression for the first order poles in terms of the zeroth order poles is in fact exactly the same.

The expression for the denominator of $Q(k)$ in Eq. (16) is written as

$$d^2S\left(k^4 + (k^2 + \kappa_t^2)k^2 + \kappa_s^2\kappa_t^2\right) + \gamma_h^2 + 2\theta\gamma_i(1-C)k\,. \qquad \text{(B.1)}$$

For $\theta = 0$ this expression reduces to a second order polynomial in $u = k^2$. Therefore its poles can be solved for analytically, at the poles $u$ satisfies

$$u_\pm = -\kappa_s^2 - \frac{1}{2l_{so}^2} \pm \frac{1}{2l_{so}^2}\sqrt{1-a}\,, \qquad \text{(B.2)}$$

where $a = 4\frac{\gamma_i^2 l_{so}^4 C}{d^2 S}$. This gives the zeroth order poles $k_j$ in Eqs. (20,21).

With this the denominator for arbitrary $\theta$ can to first order in $\kappa_s/\kappa_t$ be expressed as

$$D(k) = d^2S\left(\prod_{j=1}^{4}(k - k_{j0}) + \frac{2\gamma_i\theta(1-C)}{d^2S}k\right). \qquad \text{(B.3)}$$

The first order correction in $\theta$ to the poles can be found by substituting $k_j = k_{j0} + \theta \delta k_j$. Substituting this in Eq. (B.3) and keeping only terms up to first order in $\theta$ it follows that the pole of the new expression satisfies

$$0 = \left(\prod_{l \neq j}(k_{j0} - k_{l0})\right)\delta k_j + \frac{2\theta\gamma_i(1-C)}{d^2 S}k_{j0}. \tag{B.4}$$

Since we expand in $\theta$ this is a first order polynomial in $\delta k_j$, and hence is solved directly by

$$\delta k_j = -\frac{2\gamma_i(1-C)}{d^2 S}k_{j0}\prod_{l \neq j}\frac{1}{k_{j0} - k_{l0}}. \tag{B.5}$$

To simplify this expression further we may exploit that the zeroth order poles come in pairs with opposite sign. Using this we may simplify Eq. (B.5) to, using indices modulo 2.

$$\delta k_j = \frac{\gamma_i(1-C)}{d^2 S}\frac{1}{k_{(j+1)0}^2 - k_{j0}^2}, \tag{B.6}$$

which implies Eq. (25) in the main body.

The procedure above only holds as long as $|k_{10} - k_2| \gg |k_{1,2}|$, that is, if $|a-1| \gg \theta$. If this constraint is not satisfied we find that $|k_{10} - k_{20}|$ is of order $\theta$ or smaller as well, and hence the expansion to first order in $\theta$ is incorrect. In fact, Eq. (B.6) incorrectly predicts that $\delta k_j$ diverges as $a \to 1$.

This means that we need to expand Eq. (B.4) up to second order in this limit. We may write $k_{1,2} = k_{\pm} + \delta_{1,2}$

$$k_{\pm} = \pm i\sqrt{\kappa_s^2 + \frac{1}{2l_{so}^2}}, \tag{B.7}$$

$$\delta_j \ll k_{\pm}, \quad j = 1, 2,$$

where the appropriate sign is determined by the convergence of the kernel in the plane in which the contour is closed. Using that $k - k_1 = k_+ + \delta_{1,2} - k_1 = \delta_{1,2} + \frac{1}{2}(k_2 - k_1)$ and $k - k_2 = k_+ + \delta_{1,2} - k_2 = \delta_{1,2} - \frac{1}{2}(k_2 - k_1)$ the poles satisfy

$$0 = (k_+ - k_-)^2(\delta_{1,2}^2 - (k_1 - k_2)) + \frac{2\theta\gamma_i(1-C)}{d^2 S}k_+. \tag{B.8}$$

This is a second order polynomial in $\delta_{1,2}$. Its solutions are given by

$$\delta_{1,2} = \pm\frac{1}{2}\sqrt{(k_1 - k_2)^2 - 4\frac{2\theta\gamma_i(1-C)}{d^2 S}\frac{k_+}{(k_+ - k_-)^2}}, \tag{B.9}$$

where the $+$ sign is used for $\delta_1$ and the $-$ sign for $\delta_2$, In summary, if $|a-1| \gg \theta$ the expression for the poles in Eq. (B.6) should be used, while for $|a-1| \ll \theta$ the poles can be found using Eq. (B.9).

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
