# Peer review of "Nonreciprocal superconducting transport and the spin Hall effect in gyrotropic structures"

_SciPost Physics, doi:SciPost Phys. 16, 055 (2024)_

## Round 2 · Referee Report · Anonymous (Referee 1) · 2023-11-9

Strengths

1- The authors investigate theoretically the anomalous supercurrent and the diode effect in a hybrid gyrotropic structure consisting of a lateral Josephson junction in which the normal layer exhibits spin-orbit coupling (SOC), proximized by a ferromagnetic insulator (FI). This is a subject of current interest in the community of mesoscopic superconductivity. 2- By using the quasi-classical Usadel equation formalism, the authors show that an anomalous supercurrent arises when both the spin Hall angle (produced by SOC) and the interfacial exchange field (produced by the FI) are finite. 2- Analytic expressions for the first harmonics of the supercurrent and the anomalous phase are obtained in some limiting cases.

Weaknesses

1- In the introduction, it is not clear why the authors give such an emphasis on the concept of gyrotropy, which is never mentioned in all other sections of the paper (apart from the conclusions). 2- I think the introduction is not so clear on gyrotropy .

Report

In this paper the authors investigate theoretically the anomalous supercurrent and the diode effect in a hybrid gyrotropic structure consisting of a lateral Josephson junction in which the normal layer exhibits spin-orbit coupling (SOC), proximized by a ferromagnetic insulator (FI). By using the quasi-classical Usadel equation formalism, they show that an anomalous supercurrent arises when both the spin Hall angle (produced by SOC) and the interfacial exchange field (produced by the FI) are finite. In particular, by linearizing the boundary equations, the first harmonics of the supercurrent and the anomalous phase are obtained. Analytic expressions for such quantities are obtained in some limiting cases, while in more general situations they are numerically calculated as functions of the various parameters. To study the diode effect the authors consider the simpler system which consists of a trilayer. They find that the diode effect arises again when both the spin Hall angle and the interfacial exchange field are finite. The diode efficiency is calculated numerically for some choice of parameters. The paper is closed by a thorough discussion.

The paper presents interesting and novel results, it is well written and organized. In my opinion the paper deserves publication in SciPost Physics once the requested changes are implemented.

Requested changes

1- In the introduction, in order to improve the readability, the authors should define more clearly the meaning of gyrotropy, what gyrotropic materials are (possibly making some examples), and define the gyrotropic symmetry.

2- In Sec. II, the authors should specify how the spin Hall angle and the spin relaxation time are related to the strength of SOC. How do they depend on each other?

3- In Figs. 6, 7 and 9(b), in which units the current I(k) is plotted?

  • validity: top
  • significance: high
  • originality: high
  • clarity: high
  • formatting: excellent
  • grammar: excellent

Author:  Tim Kokkeler  on 2024-01-17  [id 4251]

(in reply to Report 1 on 2023-11-09)

We thank the Referee for their useful review, their appreciation of our analytic results and suggestions for improvement. See below for a point by point reply to their questions.

Q1 - In the introduction, in order to improve the readability, the authors should define more clearly the meaning of gyrotropy, what gyrotropic materials are (possibly making some examples), and define the gyrotropic symmetry.

A1 - We have restructured and substantially reworked the introduction so that the notion of gyrotropy is more naturally introduced, and its importance for our theory is highlighted. In particular, we have added the definition of gyrotropy, and explained the origin of terminology that goes back to crystal optics. We also refer to some explicit examples of gyrotropic materials currently used in proposals for superconducting diodes. Next to this, to connect the introduction better to the subsequent sections of the manuscript, we added connections to the concepts of gyrotropy in the sections on theory and results. It is consistently emphasized that, for the hybrid junction we considered, gyrotropy is defined by the mere existence of a bilayer structure formed by two different materials, not by microscopic symmetry breaking.

Q2 - In Sec. II, the authors should specify how the spin Hall angle and the spin relaxation time are related to the strength of SOC. How do they depend on each other?

A2 - We agree with the referee that for specific types of spin-orbit coupling the spin-Hall angle and the relaxation time are not independent. However, this relation between the spin-Hall angle and the relaxation time depends on the type of SOC (extrinsic, intrinsic, etc.), and the dominating mechanics of spin-to-charge conversion. If several microscopic mechanisms are equally relevant, such a relation can vary quite significantly. Our theory, based on kinetic equations in the diffusive limit, is general and valid for all types of SOC allowed by symmetry in macroscopically isotropic systems. In other words, it does not depend on the {microscopic origin of SOC and the specific mechanism of spin-to-charge conversion}. Therefore, θ,χ and 1/τ_so can be assumed as independent parameters. For specific types of SOC the relations between these three parameters are fixed. We added a discussion on this matter to our paper on page 4, in the paragraph preceding Eq. (4).

Q3 - In Figs. 6, 7 and 9(b), in which units the current I(k) is plotted?

A3 - We thank the referee for pointing this out, we have adapted the labels of these figures.

---

## Round 2 · Referee Report · Anonymous (Referee 2) · 2023-11-19

Report

The submitted work makes predictions on the anomalous and diode effects in a lateral Josephson junction through a thin layer of material displaying spin Hall effect, when it is deposited on top of a ferromagnetic insulator. The novelty of the study is to consider a centro-symmetric material in which the spin Hall effect is attributed to an extrinsic mechanism of spin-orbit coupling induced by the disorder. To do so, the authors make use of a recent generalization of the Usadel equations, which describe the superconducting proximity effect in the diffusive regime. The results are obtained through a detailed analytic resolution of the two-dimensional equations that need to be solved in the considered lateral geometry.

I have several remarks on the way the results are presented. Namely,

1/ The Introduction provides a description on the anomalous and diode effects on a much too general level. At the same time, it fails in positioning the novelty of the results compared with those obtained in earlier literature.

2/ It is not clear from the paper that the anomalous and diode effects have already been analyzed in centro-symmetric materials. For instance, there have been several works on Josephson junctions across 2D or 3D quantum spin Hall insulators, in which such effects are present.

3/ The results on the anomalous Josephson effect are neatly captured by Eqs. (27) and (28). At the same time, several features therein are common to the results predicted in diffusive junctions in which the spin Hall effect is of intrinsic, rather than extrinsic, origin. A detailed comparison of the similarities and differences would be much useful.

4/ The section on the diode effect is hard to follow. It seems much related to the prediction of a supercurrent flow induced in a material displaying spin Hall effect when it is coupled to a superconductor and a ferromagnet. Furthermore, the link with the lateral junction geometry considered for the anomalous effect is not immediate. It also raises the question whether the authors should have considered a full numerical solution of the equations in order to address the anomalous and diode effects at the same time.

For all these reasons, I believe that the authors should consider improving substantially the presentation of the manuscript in order to make their results more accessible.

  • validity: -
  • significance: -
  • originality: -
  • clarity: -
  • formatting: -
  • grammar: -

Author:  Tim Kokkeler  on 2024-01-17  [id 4252]

(in reply to Report 2 on 2023-11-19)

We thank the referee for their careful analysis of our manuscript, the appreciation of our analytical results and their suggestions. Below we provide a point by point reply to their remarks and questions.

Q1 - The Introduction provides a description on the anomalous and diode effects on a much too general level. At the same time, it fails in positioning the novelty of the results compared with those obtained in earlier literature.

A1 - The reason for giving an extensive discussion on gyrotropy in our introduction is twofold. On the one hand, to explain this concept originated from the field of crystal optics. On the other hand, is to emphasize that gyrotropy, and not inversion symmetry breaking, is the key to the appearance of non-reciprocal superconducting transport phenomena.
Moreover, we wanted to highlight the inherent difference between spin-charge conversion in the form of the spin-Hall effect, and spin-galvanic effect and to explain how, despite this, we may connect the two concepts using the junction geometry.
In our first attempt to make all these connections, we may not have sufficiently emphasized the novelty of our work. In this revised version, we have restructured the introduction to highlight this aspect better. In essence, our study is the first work on superconducting SGE and nonreciprocal transport effects in a system composed of spatially centrosymmetric materials in which gyrotropy emerges at the mesoscopic scale within this system. Specifically, the system we study is a lateral structure composed of heavy metal on a magnetic insulator – a combination of materials well-studied in spintronics. The upper interface of the metal is connected to two superconducting electrodes, forming a lateral Josephson junction. For the description of transport in this junction, we present a theoretical framework that unifies and generalizes two well-established theories: the Superconducting Proximity Effect and charge-spin conversion in Spin Hall systems, including phenomena like spin Hall magnetoresistance. This is another novelty in our work. Next to this, we found a geometry in which we are able to obtain the diode effect using analytic expressions, rather than relying on numerics. To make these points clear, We have emphasized the novelty of our work in our revised introduction.

Q2 - It is not clear from the paper that the anomalous and diode effects have already been analyzed in centro-symmetric materials. For instance, there have been several works on Josephson junctions across 2D or 3D quantum spin Hall insulators, in which such effects are present.

A2 - To the best of our knowledge, non-reciprocal transport in centrosymmetric materials, as reported so far, happens only at the edges or interfaces of such materials, at which the inversion symmetry of the bulk is broken on a microscopic level, via the appearance of localized surface or interface states. For example, in the case of quantum spin Hall insulators the anomalous nonreciprocal transport goes through the topologically protected edge/surface states, which, by their nature, are not inversion symmetric (in fact, gyrotropic). On the other hand, our theory does not require any topological or non-topological surface/edge states, or other types of inversion symmetry breaking at the microscopic level. We discuss {the physical mechanism of anomalous non-reciprocal transport, such as ϕ_0 and diode effects, which is present in} any metal with a nonzero spin-Hall angle, and the gyrotropy appearing on a mesoscopic, rather than microscopic level. That is, not the specifics of the edge are important, but only its mere existence. In our theory, nonreciprocal transport effects arise from spin-charge interconversion in the bulk of a centrosymmetric material. Therefore, our results show that non-reciprocal transport can be found using a far broader class of materials, since the spin-Hall angle is generally nonzero in any metal. We have added a clarification of this point and the distinction from topological materials in the introduction.

Q3 - The results on the anomalous Josephson effect are neatly captured by Eqs. (27) and (28). At the same time, several features therein are common to the results predicted in diffusive junctions in which the spin Hall effect is of intrinsic, rather than extrinsic, origin. A detailed comparison of the similarities and differences would be much useful.

A3 - We agree that there are many features that are similar for intrinsic and extrinsic types of SOC, such as the linear dependence on the strength of the exchange field and the strength of spin-orbit coupling. A new feature of our formula is that it includes, {for the first time at the analytic level}, the effect of spin-orbit relaxation in the material. In the geometry studied spin-orbit relaxation only suppresses the ϕ_0 -effect if l_so is smaller than the thickness, rather than the length of the junction. Therefore, the ϕ_0-effect is relatively robust against spin-orbit relaxation. Next to this, in our expressions we explicitly use the spin-mixing conductances G_i,G_r from SMR theory, which quantifies the anomalous superconducting transport in terms of the parameters commonly used to characterize classical spintronics effects. We have added a discussion about these points immediately after the introduction of Eq. (28).

Q4 - The section on the diode effect is hard to follow. It seems much related to the prediction of a supercurrent flow induced in a material displaying spin Hall effect when it is coupled to a superconductor and a ferromagnet. Furthermore, the link with the lateral junction geometry considered for the anomalous effect is not immediate. It also raises the question whether the authors should have considered a full numerical solution of the equations in order to address the anomalous and diode effects at the same time.

A4 - We agree that a full numerical solution of the problem would be interesting to obtain quantitative results for the diode efficiency and to find optimal parameters to enhance this efficiency. In this paper, however, it is not our main objective to optimize the diode effect, but rather to explain that diode effects and anomalous currents can be obtained even in relatively simple geometries with isotropic materials.
So far, the diode effect could only be calculated numerically. Here, we have identified a specific geometry in which one can analytically prove the existence of a diode effect. In our opinion, such analytic expressions are more illustrative, shedding light on how the parameters of the model contribute to the effect. The close connection between the two geometries and the expected effects in both can be observed, as both are described using the same kernel, Eq. (16). We have now emphasized these points in both the introduction and the conclusion.

---

## Round 3 · Referee Report · Anonymous (Referee 1) · 2024-1-24

Report

The authors have replied convinsingly to the points raised by the referees, and they have accordingly modified the manuscript.

---

## Round 3 · Author Response

Dear Editors,

We would like to thank both referees for their careful analyses, their appreciation of our analytical approach and the suggestions that we could use to improve our paper. We have substantially rewritten the introduction to emphasize better the importance of gyrotropy and its connection to our work, as well to illuminate better the novelty of our work. We also made modifications in the rest of their work based on their comments. Below, we provide a point by point response to their comments.

Yours Sincerely,
Tim Kokkeler, Ilya Tokatly, Sebastian Bergeret

---

## Round 3 · List of Changes

Report 1
In the introduction, in order to improve the readability, the authors should define more clearly the meaning of gyrotropy, what gyrotropic materials are (possibly making some examples), and define the gyrotropic symmetry.

We have restructured and substantially reworked the introduction so that the notion of gyrotropy is more naturally introduced, and its importance for our theory is highlighted. In particular, we have added the definition of gyrotropy, and explained the origin of terminology that goes back to crystal optics. We also refer to some explicit examples of gyrotropic materials currently used in proposals for superconducting diodes. Next to this, to connect the introduction better to the subsequent sections of the manuscript, we added connections to the concepts of gyrotropy in the sections on theory and results. It is consistently emphasized that, for the hybrid junction we considered, gyrotropy is defined by the mere existence of a bilayer structure formed by two different materials, not by microscopic symmetry breaking.

In Sec. II, the authors should specify how the spin Hall angle and the spin relaxation time are related to the strength of SOC. How do they depend on each other?

We agree with the referee that for specific types of spin-orbit coupling the spin-Hall angle and the relaxation time are not independent. However, this relation between the spin-Hall angle and the relaxation time depends on the type of SOC (extrinsic, intrinsic, etc.), and the dominating mechanics of spin-to-charge conversion. If several microscopic mechanisms are equally relevant, such a relation can vary quite significantly. Our theory, based on kinetic equations in the diffusive limit, is general and valid for all types of SOC allowed by symmetry in macroscopically isotropic systems. In other words, it does not depend on the {microscopic origin of SOC and the specific mechanism of spin-to-charge conversion}. Therefore, θ,χ and 1/τ_so can be assumed as independent parameters. For specific types of SOC the relations between these three parameters are fixed. We added a discussion on this matter to our paper on page 4, in the paragraph preceding Eq. (4).

In Figs. 6, 7 and 9(b), in which units the current I(k) is plotted?

We thank the referee for pointing this out, we have adapted the labels of these figures.

Report 2
The Introduction provides a description on the anomalous and diode effects on a much too general level. At the same time, it fails in positioning the novelty of the results compared with those obtained in earlier literature.

The reason for giving an extensive discussion on gyrotropy in our introduction is twofold. On the one hand, to explain this concept originated from the field of crystal optics. On the other hand, is to emphasize that gyrotropy, and not inversion symmetry breaking, is the key to the appearance of non-reciprocal superconducting transport phenomena.
Moreover, we wanted to highlight the inherent difference between spin-charge conversion in the form of the spin-Hall effect, and spin-galvanic effect and to explain how, despite this, we may connect the two concepts using the junction geometry.
In our first attempt to make all these connections, we may not have sufficiently emphasized the novelty of our work. In this revised version, we have restructured the introduction to highlight this aspect better. In essence, our study is the first work on superconducting SGE and nonreciprocal transport effects in a system composed of spatially centrosymmetric materials in which gyrotropy emerges at the mesoscopic scale within this system. Specifically, the system we study is a lateral structure composed of heavy metal on a magnetic insulator – a combination of materials well-studied in spintronics. The upper interface of the metal is connected to two superconducting electrodes, forming a lateral Josephson junction. For the description of transport in this junction, we present a theoretical framework that unifies and generalizes two well-established theories: the Superconducting Proximity Effect and charge-spin conversion in Spin Hall systems, including phenomena like spin Hall magnetoresistance. This is another novelty in our work. Next to this, we found a geometry in which we are able to obtain the diode effect using analytic expressions, rather than relying on numerics. To make these points clear, We have emphasized the novelty of our work in our revised introduction.

It is not clear from the paper that the anomalous and diode effects have already been analyzed in centro-symmetric materials. For instance, there have been several works on Josephson junctions across 2D or 3D quantum spin Hall insulators, in which such effects are present.

To the best of our knowledge, non-reciprocal transport in centrosymmetric materials, as reported so far, happens only at the edges or interfaces of such materials, at which the inversion symmetry of the bulk is broken on a microscopic level, via the appearance of localized surface or interface states. For example, in the case of quantum spin Hall insulators the anomalous nonreciprocal transport goes through the topologically protected edge/surface states, which, by their nature, are not inversion symmetric (in fact, gyrotropic). On the other hand, our theory does not require any topological or non-topological surface/edge states, or other types of inversion symmetry breaking at the microscopic level. We discuss {the physical mechanism of anomalous non-reciprocal transport, such as ϕ_0 and diode effects, which is present in} any metal with a nonzero spin-Hall angle, and the gyrotropy appearing on a mesoscopic, rather than microscopic level. That is, not the specifics of the edge are important, but only its mere existence. In our theory, nonreciprocal transport effects arise from spin-charge interconversion in the bulk of a centrosymmetric material. Therefore, our results show that non-reciprocal transport can be found using a far broader class of materials, since the spin-Hall angle is generally nonzero in any metal. We have added a clarification of this point and the distinction from topological materials in the introduction.

The results on the anomalous Josephson effect are neatly captured by Eqs. (27) and (28). At the same time, several features therein are common to the results predicted in diffusive junctions in which the spin Hall effect is of intrinsic, rather than extrinsic, origin. A detailed comparison of the similarities and differences would be much useful.

We agree that there are many features that are similar for intrinsic and extrinsic types of SOC, such as the linear dependence on the strength of the exchange field and the strength of spin-orbit coupling. A new feature of our formula is that it includes, {for the first time at the analytic level}, the effect of spin-orbit relaxation in the material. In the geometry studied spin-orbit relaxation only suppresses the ϕ_0 -effect if l_so is smaller than the thickness, rather than the length of the junction. Therefore, the ϕ_0-effect is relatively robust against spin-orbit relaxation. Next to this, in our expressions we explicitly use the spin-mixing conductances G_i,G_r from SMR theory, which quantifies the anomalous superconducting transport in terms of the parameters commonly used to characterize classical spintronics effects. We have added a discussion about these points immediately after the introduction of Eq. (28).

The section on the diode effect is hard to follow. It seems much related to the prediction of a supercurrent flow induced in a material displaying spin Hall effect when it is coupled to a superconductor and a ferromagnet. Furthermore, the link with the lateral junction geometry considered for the anomalous effect is not immediate. It also raises the question whether the authors should have considered a full numerical solution of the equations in order to address the anomalous and diode effects at the same time.

We agree that a full numerical solution of the problem would be interesting to obtain quantitative results for the diode efficiency and to find optimal parameters to enhance this efficiency. In this paper, however, it is not our main objective to optimize the diode effect, but rather to explain that diode effects and anomalous currents can be obtained even in relatively simple geometries with isotropic materials.
So far, the diode effect could only be calculated numerically. Here, we have identified a specific geometry in which one can analytically prove the existence of a diode effect. In our opinion, such analytic expressions are more illustrative, shedding light on how the parameters of the model contribute to the effect. The close connection between the two geometries and the expected effects in both can be observed, as both are described using the same kernel, Eq. (16). We have now emphasized these points in both the introduction and the conclusion.

---

## Editorial Decision

published